# Near-Miss Bi-Homogenous Symmetric Polyhedral Cages

**Bernard Piette \*** 🆔 **and Árpad Lukács** 🆔

Department of Mathematical Sciences, Durham University, Durham DH1 3LE, UK; arpad.l.lukacs@durham.ac.uk
\* Correspondence: b.m.a.g.piette@durham.ac.uk

**Abstract:** Following the discovery of an artificial protein cage with a paradoxical geometry, we extend the concept of homogeneous symmetric congruent equivalent near-miss polyhedral cages, for which all the faces are equivalent, and define bi-homogeneous symmetric polyhedral cages made of two different types of faces, where all the faces of a given type are equivalent. We parametrise the possible connectivity configurations for such cages, analytically derive p-cages that are regular, and numerically compute near-symmetric p-cages made of polygons with 6 to 18 edges and with deformation not exceeding 10%.

**Keywords:** uniform polyhedra; polyhedral cages; platonic group; near-miss cages; Cayley graph; protein cage; nanocage; nanoparticle

## 1. Introduction

Recently, an artificial protein structure, referred to as TRAP-cage, was engineered from the trp RNA-binding attenuation (TRAP) protein [1–4]. It is an 11-subunit RNA-binding protein that regulates the expression of the genes involved in tryptophan metabolism (trp) in Bacillus subtilis.

This nanocage consists of 24 nearly regular hendecagonal (polygon with 11 edges) faces, each of which has 5 neighbours with which it shares an edge. The 6 edges per face not shared with another face define the boundary of 38 holes; of these holes, 32 are triangles to which 3 faces contribute 1 edge each. The other six holes are made from the edges of four hendecagonal faces, which contribute to the holes with two of their edges. Some similar nearly regular structures made of the same protein have been recently identified [5,6].

The geometrical structure of the discovered protein cage was defined as a polyhedral cage (p-cage) in [7]. Mathematically, the p-cage corresponding to the TRAP-cage cannot be constructed with regular hendecagons; the edge lengths and angles of the hendecagonal faces must be slightly deformed relative to a regular polygon [7,8]. These objects are called near-miss p-cages.

Artificial polyhedral nanostructures are not new, and they are not restricted to proteins. A good example is given by DNA origami [9–11]. Unlike protein nanocages, these DNA structures are mostly hollow, as the DNA strands span the edges of what we call faces. This being said, the regular or nearly regular geometries identified in this study could be useful for a range of other nanostructures, such as DNA origami.

We should also point out that the concept of chemical cages is not new, and these cages have been observed or made in a number of contexts [12–16]. Moreover, in chemistry, polyhedral structures are also quite common [12,17–19].

Quite a range of artificial protein cages have recently been experimentally generated [20–25]. Interestingly, and unlike virus capsids, most of them need metal atoms to create the strong bonds required to bind the different proteins together [26]. The main aim for generating these artificial nanocages is to develop new methods for drug delivery [27–31]. The drug is encapsulated inside the cage, while specific receptors are bound to the outside of the cage to bind with the targeted cells (typically, cancer cells) [27]. Once inside the

targeted cell, the protein cage opens to release the drug in the cytoplasm of the cell [32]. As a result, only the cells that are targeted receive the drug, instead of most of the cells of the body, when the drug is injected in the blood stream. Moreover, it means that a smaller quantity of the drug is needed, thus significantly reducing the cost when the drug is a very expensive active compound.

The structure of virus capsids is essentially based on the geometry of platonic solids [33]. Some of the cages created experimentally, on the other hand, exhibit structures that are completely different [2,5,6]. This raises the question as to which are the best geometries for such cages [34–36]. In [7,8], large numbers of polyhedral cages were identified. They were constructed by requiring that all the faces of a given p-cage to be equivalent.

In this study, we extended these studies by constructing p-cages made of two families of polygons such that each face is equivalent to all the faces belonging to the same family. As the number of such potential p-cages is very large, we also restricted ourselves to p-cages where faces of a given type can only be attached to faces of the other type. Our aim was is to provide bionanoengineers with a list of geometries from which nanocages can be constructed, helping them to decide which polygonal protein structures to use to build such cages, similar to those described in [1–4].

The structure of our paper is as follows: After a few formal definitions, we recall how the planar graphs of regular solids are linked to the connectivity between the faces of equivalent p-cages. We then construct all the planar graphs made of two families of vertices such that each vertex of a given type is linked to a vertex of the other type and so that each vertex of a given family is equivalent to all the other vertices of the same family.

We then use the obtained graphs and the corresponding solids to determine the possible configurations for the corresponding p-cages; we used a computer program to determine those that have regular faces or irregular faces with deformation not exceeding 10%. We conclude by describing the obtained p-cages and by presenting the images of some regular ones as well as some of the least deformed (near-miss) ones.

## 2. Bi-Homogeneous Symmetric-1-2 Polyhedral Cages

As defined in [7,8], a polyhedral cage is an assembly of planar polygons, which we call faces, and holes, which are usually neither planar nor regular. Every edge must then either belong to two faces or to one face and a hole. The edges of the polygonal faces adjacent to another face are called shared edges, while the edges adjacent to a hole are called hole edges. We also impose the following two conditions: When two edges are adjacent to each other, at least one of them must be adjacent to a hole. Moreover, each face must have at least three neighbours. Together, these two conditions imply that the faces of the p-cages must have at least six edges.

A p-cage is said to be convex if the holes can be filled in with triangles in such a way that the resulting polyhedron is convex. In what follows, we only consider convex p-cages.

If all the faces of a p-cage are polygons with the same number of edges, the p-cage is said to be homogeneous [7]. We now define bi-homogeneous symmetric-1-2 (BiHS12) p-cages as p-cages made of two types of polygons, where all the faces of a given type are equivalent, such that for each pair of faces of a given type, there is a congruent automorphism (a proper rotation) of the p-cage that maps one of the faces onto the other. This implies that all the faces of a given type are identical.

P-cages made of regular polygons are defined as regular. If, on the other hand, the faces are slightly irregular, the p-cages are said to be a near miss. For some near-miss p-cages, the deformations are so small that they can hardly be noticed with the naked eye, while for other p-cages, they can be rather large. As a result, we define below a measure of the amount of deformation and restrict ourselves to deformations not exceeding 10%.

On any face, between the shared edges, there are some hole edges. On the planar graph corresponding to the hole polyhedron of the cage, these numbers may be added as labels $q_{i,v}$ around each vertex $v$, where there is one such label between any two edges around the vertex, so $i$ goes from 1 to the rank of vertex $v$. In the case of a bi-homogeneous

cage, the labels around each vertex corresponding to a face of type 1 or 2 are identical; so, instead of the index $v$, in $q_{i,v}$, one may use the notation $q_i$ and $Q_i$ for the two types of vertices of the graph.

In what follows, we use $N_1$ and $N_2$ to denote the number of faces of type 1 and 2 of the p-cage and $P_1$ and $P_2$ to denote the number of edges of the faces of each type. Each hole will be made of $\Omega_h$ edges where we include a hole index $h$ as a p-cage can have different types of holes.

## 3. Bi-Homogeneous-Symmetric-1-2 P-Cage Construction

As described in [7,8], joining the centres of the faces of the p-cage that share one edge generates an irregular polyhedron, which does not usually have planar faces, but the corresponding graph is a planar graph [37] (see Figure 1). We called this irregular polyhedron the hole-polyhedron because, by construction, the faces, the vertices and the edges correspond, respectively, to the holes of the p-cage, the faces of the p-cage, and the links between the p-cage faces. This corresponds to the dual of the p-cage as it clearly encapsulates the connectivity between the p-cage faces.

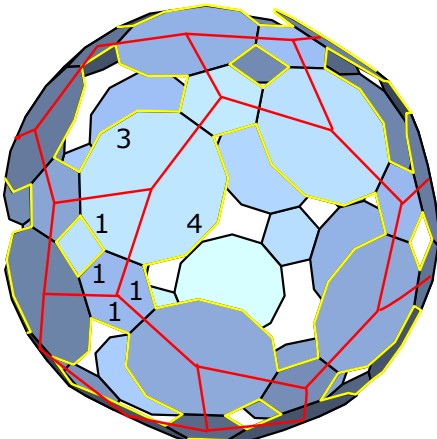

**Figure 1.** Schematic construction of the hole polyhedron of a p-cage. The number of hole–edges is written on one hexagon and one hendecagon. The hole–edges are coloured in yellow, except for the faces on the opposite side seen through the holes, while the shared edges are black.

Equivalence between the faces of the p-cage is translated onto an equivalence between the vertices of the hole–polyhedron. The planar graphs of the hole-polyhedron for bi-homogeneous-symmetric p-cages have vertices split in two sets; for any two vertices of a given type, there is an automorphism of the graph that maps one of the vertices into the other. The number of such graphs is very large, and we restrict ourselves to graphs where the vertices of a given type are linked to vertices of the other type. This is equivalent to saying that the p-cage faces of type 1 are neighbours of faces of type 2 and vice versa. We call these planar graphs BiHS12 planar graphs.

To construct a p-cage, one must first chose a BiHS12 planar graph and place polygons of type 1 onto each type 1 vertex of the BiHS12 planar graph and place polygons of type 2 onto the type 2 vertices. The hole–edges can be distributed in different ways between the corners around each vertex. For example, when placing an octagon on a trivalent vertex, such as on a tetrahedron, there are three shared edges and five hole–edges that must be distributed between the three adjacent faces of the planar graph. This can be achieved as 1,1,3 or 1,2,2, plus permutations, and this must also be performed for each face of the p-cage in such a way that the faces of the p-cage are all equivalent (see [7]).

Formally, given a BiHS12 planar graph, if a type 1 vertex, on which we place a $P_1$-gonal face for the p-cage, has $d$ neighbours, then the numbers $q_i$ ($i = 1 \ldots d$) of hole–edges on each corner around that vertex must satisfy $\sum_{i=1}^{d} q_i = P_1 - d$. Similarly, for type 2 vertices, we have $\sum_{i=1}^{d} Q_i = P_2 - d$. Requiring the faces of the p-cage be equivalent implies that the

corresponding vertices of the hole–polyhedron graph must be equivalent. This implies that the sequence $q_i$ must be identical for all the type 1 vertices up to a cyclic rotation, which is also determined by the equivalence between the hole–polyhedron vertices. Similarly, the sequence $Q_i$ must be identical for all the type 2 vertices. As we shall see, for some p-cages, the equivalence imposes that some pairs of $q_i$ and $Q_i$ must be identical.

As a first step, we characterise all the BiHS12 planar graphs.

## 4. Bi-Homogeneous-Symmetric-1-2 Planar Graphs

We denote, respectively, the number of vertices, faces, and edges of the planar graph by $V$, $F$, and $E$; and, by Euler's formula, they satisfy the constraint $V + F - E = 2$. We have vertices of 2 different numbers of edges, $L_1$ and $L_2$, and we denote the number of vertices of type $j$ by $V_j$.

If adjacent vertices are of different types, the vertices around a face of the planar graph must be of alternating types, implying that the planar graph faces must have an even number of edges and have the same number of vertices of each type. In a planar graph, each edge belongs to two faces; therefore, the number of edges can be obtained as one half of the sum of edges of the faces. Let $f_i$ denote the number of faces with $2i$ edges; with this notation,

$$E = \sum_i i f_i. \tag{1}$$

As we assumed that vertices of type 1 are only connected to those of type 2; each edge belongs to a type 1 and a type 2 vertex. Again, the number of edges is obtained as

$$E = L_1 V_1 = L_2 V_2. \tag{2}$$

As all edges have one end on a type 1 and one on a type 2 vertex, the number of vertices in any of the types can be obtained by dividing the total number of edges with the ranks of the vertices in one type,

$$V_j = \frac{E}{L_j} = \sum_i \frac{i f_i}{L_j}, \tag{3}$$

where, in the second equality, we used (1). Plugging $V = V_1 + V_2$ and (2), written in the form $E = (L_1 V_1 + L_2 V_2)/2$, into Euler's formula yields

$$F + V_1 \left(1 - \frac{L_1}{2}\right) + V_2 \left(1 - \frac{L_2}{2}\right) = 2. \tag{4}$$

The total number of faces is the sum of the number of faces with a given number of edges,

$$F = \sum_i f_i, \tag{5}$$

which, together with (4), yields

$$2 L_1 L_2 = \sum_i f_i [L_1 L_2 + i(L_1 + L_2 - L_1 L_2)]. \tag{6}$$

Moreover, if each vertex of type $j$ belongs to $v_{j,i}$ faces with $i$ vertices of each type,

$$V_j = \frac{i f_i}{v_{j,i}}. \tag{7}$$

and

$$\sum_i v_{j,i} = L_j. \tag{8}$$

One must therefore find all the $i$ and $f_i$ satisfying (6). Then, compute $V_1$ and $V_2$ using (3), which must be integers, and compute $v_{1,i}$ and $v_{2,i}$ using (7), which must also be integers that satisfy (8). We wrote a computer program that tests these conditions and solved this problem for the cases $L_1 \in [3-6]$ and $L_2 \in [L_1-6]$. After eliminating the numerical solutions that did not correspond to any planar graph, we obtained the graphs described below. The naming convention for the corresponding p-cages is similar to the one used in [8], i.e., $NAME\_Pp_1\_Pp_2\_q_1\_\ldots\_q_n - Q_1\_\ldots\_Q_N$ where $NAME$ is one of sp, Ato, Atco, Atid, DArd, or DArt for, respectively, the square prism, the truncated octahedron, the truncated cuboctahedon, the truncated icosidodecahedron, the rhombic dodecahedron (dual of the cuboctahedon), and the rhombic triacontahedron (dual of the icosidodecahedron). $p_1$ and $p_2$ are the numbers of edges for faces of type 1 and 2 respectively, while $q_i$ and $Q_i$ (see Section 2) are the number of hole edges for faces of type 1 and 2, respectively. When naming or labelling the p-cages we use the following equivalences: $a = q_1, b = q_2, c = q_3$ and $A = Q_1, B = Q_2, C = Q_3, D = Q_4, E = Q_5$.

Using our computer program, we found that the only BiHS12 planar graphs are, as shown in Figure 2, the following:

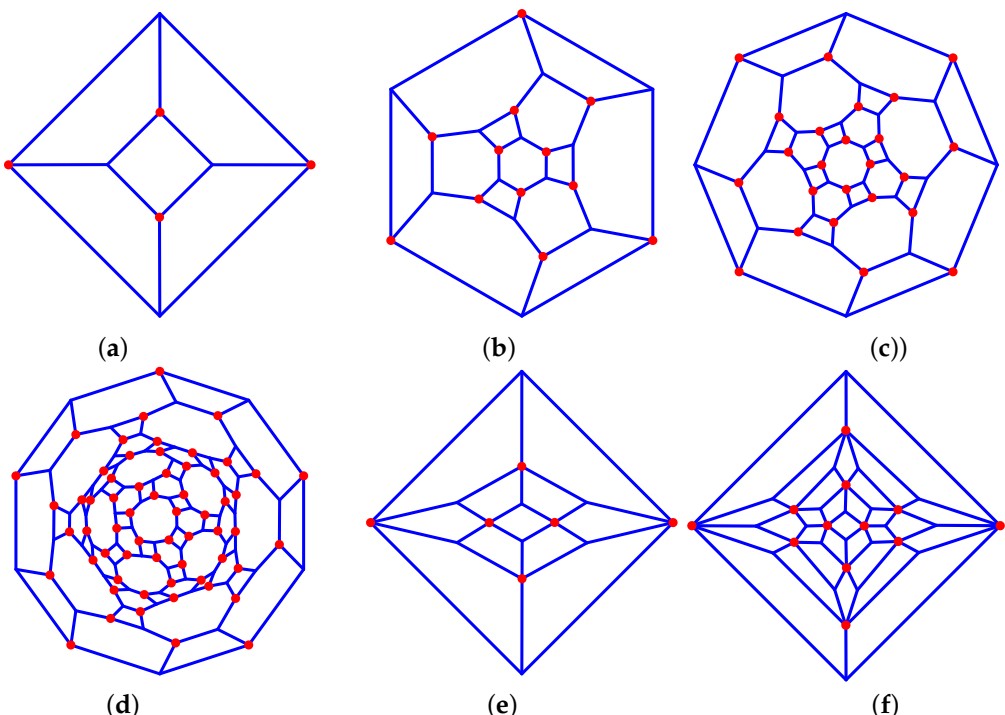

**Figure 2.** Bi-homogeneous-symmetric-1-2 planar graphs: (**a**) square prism, (**b**) truncated octahedron, (**c**) truncated cuboctahedron, (**d**) truncated icosidodecahedron, (**e**) rhombic dodecahedron, and (**f**) rhombic triacontahedron.

- The graph of any prism with a P-gon base where $P$ is even: $L_1 = L_2 = 3$, $V_1 = V_2 = P$, $n_2 = P$, $n_k = 2$, $v_{1,2} = v_{2,2} = 2$, $v_{1,k} = v_{2,k} = 1$. We only consider the square prism, as prisms with larger bases lead to cages with very large holes [8]. The p-cages are named sp_Pp1_Pp2_a_b_c-A_B_C.
- The graph of the truncated octahedron is: $L_1 = L_2 = 3$, $V_1 = V_2 = 12$, $n_2 = 6$, $n_3 = 8$, $v_{1,2} = v_{2,1} = 1$. $v_{1,k} = v_{2,k} = 2$. The p-cages are named Ato_Pp1_Pp2_a_b_c-A_B_C
- The chiral graph of the truncated cuboctahedron is: $L_1 = L_2 = 3$, $V_1 = V_2 = 26$, $n_2 = 12$, $n_3 = 8$, $n_4 = 6$, $v_{1,2} = v_{2,1} = 1$, $v_{1,3} = v_{2,3} = 1$, $v_{1,4} = v_{2,4} = 1$. The p-cages are named Atco_Pp1_Pp2_a_b_c-A_B_C.
- The graph of the truncated icosidodecahedron: $L_1 = L_2 = 3$, $V_1 = V_2 = 60$, $n_2 = 30$, $n_3 = 20$, $n_5 = 12$, $v_{1,2} = v_{2,1} = 1$, $v_{1,3} = v_{2,3} = 1$, $v_{1,5} = v_{2,5} = 1$. The p-cages are named Atid_Pp1_Pp2_a_b_c-A_B_C.

- The graph of the rhombic dodecahedron, the dual of the cuboctahedron, is: $L_1 = 3$, $L_2 = 4$, $V_1 = 8$, $V_2 = 6$, $n_2 = 12$, $v_{1,2} = 3$, $v_{2,2} = 4$. The p-cages are named `DArd_Pp1_Pp2_a_b_c-A_B_C_D`.
- The graph of the rhombic triacontahedron, the dual of the icosidodecahedron, is: $L_1 = 3$, $L_2 = 5$, $V_1 = 10$, $V_2 = 12$, $n_2 = 30$, $v_{1,2} = 3$, $v_{2,2} = 5$. The p-cages are named `DArt_Pp1_Pp2_a_b_c-A_B_C_D_E`.

## 5. Labelling of Hole–Edges

We now identify all the possible configurations for the BiHS12 p-cages before we determine the coordinate positions of all the vertices corresponding to the planar faces for the p-cages. We then select those for which the face deformation, defined below, does not exceed 10%.

In this section, we list all the planar graphs we identified as having equivalence symmetry and label the corners around each vertex of the graphs so that there is an automorphism of the graph, making all the vertices of type 1 equivalent to each other and all the vertices of type 2 equivalent to each other.

Note that swapping $a, b, c$ with $A, B, C$ corresponds to swapping $P_1$ and $P_2$ and leads to the same or the chiral p-cage, so we only need to consider p-cages with $P_2 \leq P_1$.

### 5.1. `sp_Pp1_Pp2_a_b_c-A_B_C`

As there are two vertices of a given type on the same square, at least one of them must face the same type of vertex on the same square. We label this `a`. The other two, `b` and `c`, can either face each other or face the same label. If `b` faces `c`, then `a = b = c`. The only possibility is for each label to face the same label diagonally opposite, and this applies to the two types of vertices. See Figure 3a.

Other prisms have a similar labelling with alternating labels `a` and `A` on the base of the prism, while the quadrilaterals on the sides have alternating `bBbB` and `cCcC` labels.

There are equivalences between labels. The permutation, $a \Leftrightarrow A$, $b \Leftrightarrow C$, $c \Leftrightarrow B$ corresponds to a rotation of the square prism, while $a \Leftrightarrow A$, $b \Leftrightarrow B$, $c \Leftrightarrow C$ corresponds to the chiral p-cage. Moreover, any cyclic permutation of the pairs $a, A$, $b, B$, and $c, C$ corresponds to a chiral or an identical p-cage.

### 5.2. `Ato_Pp1_Pp2_a_b_c-A_B_C`

One of the labels must be placed on the square, for which we use the letter `a`. Then, via construction, some of the hexagons are all `b`, and the others are all `c`. The same applies to both types of vertices. See Figure 3b.

If $P_1 = P_2$ and $a = A$, $b = B$, and $c = C$, then `Ato_Pp_Pp_a_b_c-a_b_c` is identical or the chiral version of `Ato_Pp_Pp_a_c_b-a_c_b`.

### 5.3. `Atco_Pp1_Pp2_a_b_c-A_B_C`

By equivalence, a given label must be placed either on a square, a hexagon, or an octagon. See Figure 3c.

### 5.4. `DArd_Pp1_Pp2_a_b_c-A_B_C_D`

By equivalence, a given label must be placed either on a square, an hexagon, or a decagon. See Figure 3d.

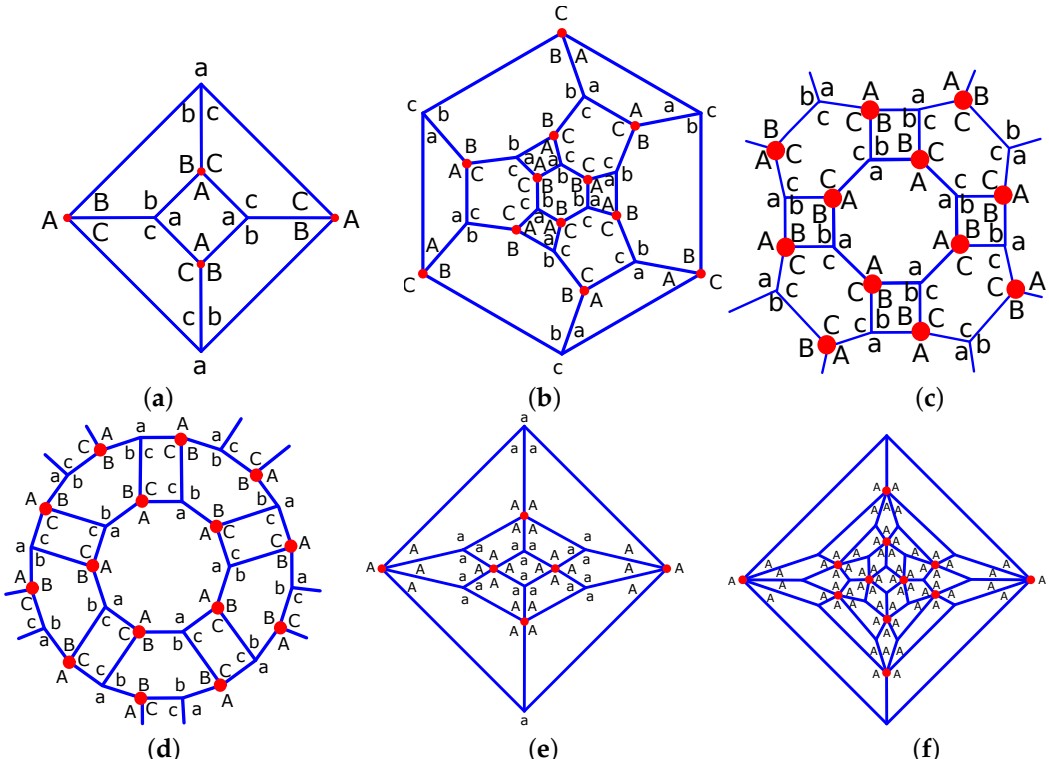

**Figure 3.** Labelling for the (**a**) cube, (**b**) truncated octahedron, (**c**) truncated cuboctahedron, (**d**) truncated icosidodecahedron, (**e**) rhombic dodecahedron, and (**f**) rhombic triacontahedron. (The labels on type 1 vertices are not shown.)

### 5.5. DArd_Pp1_Pp2_a_b_c-A_B_C_D

The symmetry group of the rhombic dodecahedron is the symmetry group of the cube. For the two types of faces to be equivalent, one must find a subgroup that acts transitively on each set of vertices. This means that we have to find a subgroup for which the order is a multiple of 6 and 8, and this corresponds to the full symmetry group of the cube, which is of order 24 [8,37–39]. Applying that symmetry, we easily see that the only equivalent configuration is to have a = b = c and A = B = C = D. See Figure 3e.

### 5.6. DArt_Pp1_Pp2_a_b_c-A_B_C_D_E

The symmetry group of the rhombic triacontahedron is the symmetry group of the dodecahedron. For the two types of faces to be equivalent, one must find a subgroup that acts transitively on each set of vertices. This means that we have to find a subgroup for which the order is a multiple of 10 and 12; this corresponds to the full symmetry group of the dodecahedron that is of order 60 [8,37–39]. Applying that symmetry, we easily see that the only equivalent configuration is to have a = b = c and A = B = C = D = E. See Figure 3f.

## 6. Optimisation

Having identified all the possible configurations for the p-cage, we proceed by defining a measure for the deviations of the faces from regular polygons. We also define a function that measures the irregularity of the p-cage so that we can later use a simulated annealing method to determine the configurations that are regular or the least irregular.

For a p-cage to be regular, all the faces, all the edges, and all the angles of the polygonal faces must be identical. For a $P$-gon, this means that all the edges must have the same length $L$ and the same angle $\pi(1 - 2/P)$. Near-miss p-cages are p-cages where the faces are not regular polygons but close to being regular. Irregular faces have edge lengths and angles slightly different from those of a regular one.

To evaluate the level of regularity of the p-cage, we first determine the distance $d_i$ between vertices $i$ and $i + 1$ as well as the angle $\alpha_i$ between the segments $(i - 1, i)$ and

$(i, i+1)$. Defining the following energy function to evaluate the deformation of each type of face

$$E_j = \sum_{i=1}^{P_j} \left[ c_l \left( \frac{d_i - L}{L} \right)^2 + c_a \left( \frac{\alpha_i - \pi(1 - \frac{2}{P_j})}{\pi(1 - \frac{2}{P_j})} \right)^2 \right], \tag{9}$$

the function we have to minimise is

$$E = \frac{1}{P_1 + P_2} \left( E_1 + E_2 + c_c\, E_{\text{Fconv}} + c_{pc}\, E_{\text{Pconv}} \right) \tag{10}$$

where $c_l$, $c_a$, and $c_c$ are three weight factors. $E_{\text{Fconv}}$, given explicitly by (12), is 0 unless the polygon defined by the vertices is concave. $E_{\text{Pconv}}$, given explicitly by (13), is 0 unless the p-cage is concave. These last two terms were used in the simulated annealing to enforce the convexity of the faces and the p-cage by taking large values for $c_c$ and $c_{pc}$.

We divide the sum by $P$ to approximately set the same energy scale for each $P$. This facilitates the parametrisation of the optimising algorithm.

We consider all p-cages with $P = 6$ to 18. As large values of $q$ and $Q$ lead to very large holes, we restrict ourselves to values of $q_i$ and $Q_i$ taking values between 1 to 5.

To characterise the face, with normal vector $m_f$, we define $n_i$ as its vertices, ordered anticlockwise. Then, to measure the angle $\alpha_i$ and edge length $d_i$, we define $v_i = n_i - n_{i-1}$, evaluate $v_i \times v_{i+1}$, and

$$\text{if } (v_i \times v_{i+1}) \cdot m_f \geq 0 \quad : \quad \alpha_i = \pi - \text{acos}(\frac{(v_i \cdot v_{i+1})}{|v_i||v_{i+1}|}), \quad d_i = |v_i|,$$

$$\text{if } (v_i \times v_{i+1}) \cdot m_f < 0 \quad : \quad \alpha_i = \pi + \text{acos}(\frac{(v_i \cdot v_{i+1})}{|v_i||v_{i+1}|}), \quad d_i = |v_i|. \tag{11}$$

Note that $\alpha_i$ in (11) corresponds to the angle inside the face, which is larger than $\pi$ if the face is not convex. If $m_f$ is the vector normal to the face and if $n_i$ are running anticlockwise when seeing the face in the direction of $n_f$, then, using the Heaviside function $H(x)$,

$$E_{\text{Fconv}} = \frac{1}{P} \sum_i \left[ H\left( (v_i \times v_{i+1}) \cdot m_f \right) \right]. \tag{12}$$

If $V_i$ is the position of the centre of face $i$ and if we consider two adjacent faces $V_i$ and $V_j$ with normal vectors $m_i$ and $m_j$, respectively, we can check if the p-cage is convex by computing the distance between the centres of the two faces as well as the distance between $V_i + m_i$ and $V_j + m_j$. If the latter is the largest for all pairs of adjacent faces, then the p-cage is convex. We can then use the following expression for $E_{\text{Pconv}}$:

$$E_{\text{Pconv}} = \frac{1}{P} \sum_i \left[ H\left( |V_i - V_j|^2 - |V_i + m_i - V_j + m_j|^2 \right) \right]. \tag{13}$$

We define the length and angle deformations as follows:

- Length : $\Delta_l = \max_i \left( \left| \frac{d_i - L}{L} \right| \right)$

- Angle : $\Delta_a = \max_i \left( \left| \frac{\alpha_i - \pi(1 - \frac{2}{P})}{\pi(1 - \frac{2}{P})} \right| \right)$

In most cases, near-miss p-cages can be deformed smoothly, changing the edge lengths as well as the angles and, as a result, both $\Delta_l$ and $\Delta_a$. Identifying near-miss p-cages for a given connectivity (fixed hole–polyhedron, $P_1$, $P_2$, $q_i$, and $Q_i$) consists of finding the geometry that minimises $\Delta_l$ and $\Delta_a$. This can be achieved by minimising the function (10) over the coordinates of the vertices. As in [7], we performed this using a simulated annealing algorithm, for a range of values of $c_l$ and $c_a$ satisfying the constraint $c_l + c_a = 2$.

After removing those with crossing faces from the obtained p-cages, we selected the configuration with the smallest deformation, i.e., those for which the maximum value of $\Delta_l$ and $\Delta_a$ was the smallest.

Some regular convex p-cages ($\Delta_l = \Delta_a = 0$) for $P_1 = P_2$ were already derived analytically in [7].

## 7. Parametrisation

In order to parameterise the p-cages, we introduce the following parametrisation of the planes containing the reference faces

$$\mathcal{P}_1(t_1, t_2) = V_i + t_1 v_{i1} + t_2 v_{i2}, \qquad \mathcal{P}_2(s_1, s_2) = W_i + s_1 w_{i1} + s_2 v_{i2}, \tag{14}$$

where $i$ is the index of the faces adjacent to the first plane, while $t_1, t_2, s_1$, and $s_2$ are parameters. $V$ and $W$ are arbitrary vectors, and the plane basis vectors $v_{i1}$ and $v_{i2}$ can be assumed to be orthonormal, similarly for $w_{i1}$ and $w_{i2}$.

For near-miss p-cages, the vectors $V$ and $W$ are adjusted using a simulated annealing procedure, while their orientations are constrained by the symmetry of the p-cage. As a starting point, we consider that $V_0$ and $W_0$ point to two adjacent vertices of the hole–polyhedron.

We then chose some simple vectors: $v_0$ for the first plane of type 1 and $w_0$ for the first plane of type 2, and use

$$v_{11} = \left( \mathbb{1} - \frac{V_1 V_1^t}{|V_1|^2} \right) v_0, \qquad\qquad v_{12} = \frac{V_1 \times v_{11}}{|V_1 \times v_{11}|^2},$$

$$w_{11} = \left( \mathbb{1} - \frac{W_1 W_1^t}{|W_1|^2} \right) w_0, \qquad\qquad w_{12} = \frac{W_1 \times w_{11}}{|W_1 \times w_{11}|^2}. \tag{15}$$

The vectors spanning the other faces are obtained using the symmetry of the p-cage.

One of the problems we have to solve is to find the intersection between these two planes. First, we define the normal vectors, $p$ and $q$, to the planes as well as the vector $u$ parallel to the plane intersection:

$$p = v_1 \times v_2, \qquad q = w_1 \times w_2, \qquad u = q \times p. \tag{16}$$

In order to fix a specific point on the line of intersection, we choose the one that is perpendicular to $u$. We then have

$$U = V + t_1 v_1 + t_2 v_2 = W + s_1 w_1 + s_2 w_2 \tag{17}$$

and multiplying (17) by $u$ leads to a relationship between $t_1$ and $t_2$ as well as between $s_1$ and $s_2$. Now, multiplying (17) by $q$, we obtain an expression for $t_1$ that, as detailed in [8], when inserted back into (17), gives

$$U = V + \frac{(q \cdot (W - V))(u \cdot v_2) + (u \cdot V)(q \cdot v_2)}{(q \cdot v_1)(u \cdot v_2) - (u \cdot v_1)(q \cdot v_2)} \left( v_1 - \frac{(u \cdot v_1)}{(u \cdot v_2)} v_2 \right) - \frac{(u \cdot V)}{(u \cdot v_2)} v_2. \tag{18}$$

We are now ready to construct each family of p-cages one by one by considering their specific symmetries.

### 7.1. Square Prism

The vertices of the square prism that are opposite to each other on the prism face span two not necessarily regular tetrahedra on which the faces of both types are placed. These two tetrahedra can then be rotated with respect to each other.

The faces of the p-cage are placed around the corners of the square prism. The corners of a cube are at $(\pm 1, \pm 1, \pm 1)$, but as the square prism can be elongated or squashed

vertically, we have an extra degree of freedom, as we can rotate the vector $(1,1,1)$ by an angle $\theta$ around the axis

$$g_0 = (-1,1,0). \tag{19}$$

We consider (Figure 4) the following vectors for the normal to the face of type 1 ($V_i$) and type 2 ($W_i$):

$$
\begin{aligned}
V_1 &= (x_1,y_1,z_1)^t, & V_2 &= R_z(\pi)\,V_1, & V_3 &= R_x(\pi)\,V_1, & V_4 &= R_y(\pi)\,V_1, \\
W_1 &= (x_2,y_2,z_2)^t, & W_2 &= R_z(\pi)\,W_1, & W_3 &= R_y(\pi)\,W_1, & W_4 &= R_z(\pi)\,W_1.
\end{aligned} \tag{20}
$$

We use $V_1$ as the reference frame. The base vectors for the reference faces are given by (15), where

$$v_0 = (-1,1,0)^t, \qquad w_0 = (-1,-1,0)^t. \tag{21}$$

The bases for the other faces are obtained from $v_{11}, v_{12}$ or $w_{11}, w_{12}$ by applying the rotations relating the corresponding faces to the reference faces as described in (20).

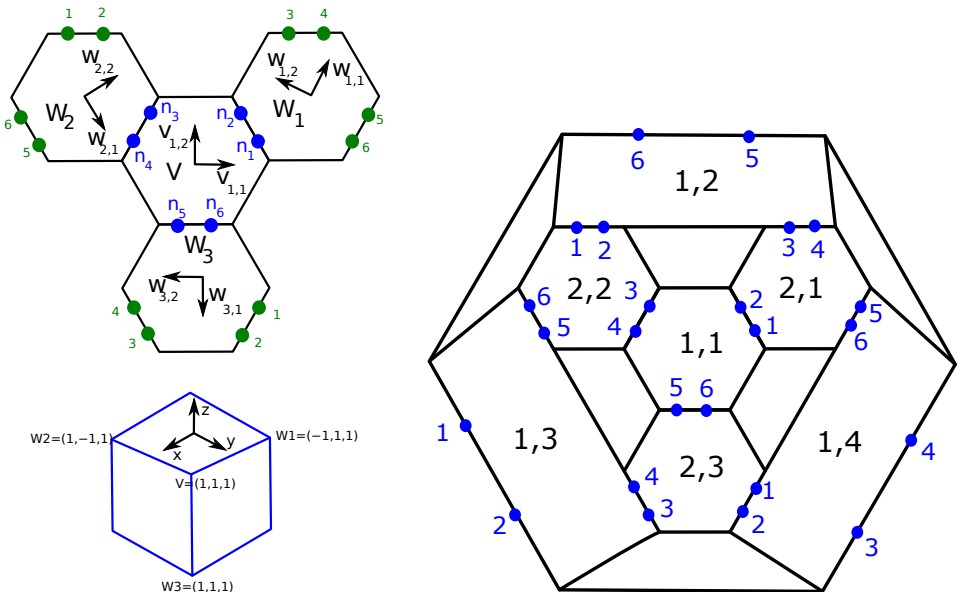

**Figure 4.** Labelling the vertices of the reference faces of the square prism p-cage. The letter $v$ is used for the type 1 faces while the letter $w$ is used for the type 2 faces. The vertices, labbeled $n$ are numbered anti-clockwise on the face of type 1 and clockwise on the face of type 2.

We use $V, v_{11}, v_{12}, W_i, w_{i,1}$ and $w_{i,2}$ for $i = 1, 2, 3$ in (18) and (16) to compute the vectors spanning the line of intersection between adjacent faces $U_i$ and $u_i$. Then, the vertices on the face intersection are

$$
\begin{aligned}
n_1 &= U_1 + t_1 u_1, & n_2 &= U_1 + t_2 u_1, \\
n_3 &= U_2 + t_3 u_2, & n_4 &= U_2 + t_3 u_2, \\
n_5 &= U_3 + t_5 u_3, & n_6 &= U_3 + t_6 u_3.
\end{aligned} \tag{22}
$$

The optimisation parameters are $V_1$, $W_1$, $t_1, t_2, t_3, T_4, t_5, t_6$, from Equation (22), as well as the coordinates, in the face plane, of the nonshared vertices.

Given the faces constructed around $V_1$ and $W_1$, we can construct the remaining six faces by applying the rotations $R_z(\pi)$, $R_y(\pi)$, and $R_x(\pi)$ to these reference faces.

7.1.1. Regular P-Cages

For the cube

$$V = S_1(1,1,1), \qquad\qquad W = S_2(-1,1,1), \qquad\qquad (23)$$

$S_1$ and $S_2$ are two scaling parameters. Then,

$$v_{11} = \frac{1}{\sqrt{2}}(-1,1,0), \qquad v_{12} = \frac{1}{\sqrt{6}}(-1,-1,2),$$

$$w_{11} = \frac{1}{\sqrt{2}}(-1-,1,0), \qquad w_{12} = \frac{1}{\sqrt{6}}(1,-1,2),$$

$$p = (1,1,1)\frac{1}{\sqrt{3}}, \qquad q = (-1,1,1)\frac{1}{\sqrt{3}}, \qquad u = \left(0,\frac{2}{3},-\frac{2}{3}\right), \qquad (24)$$

$$(u \cdot V) = 0, \qquad (u \cdot v_{11}) = \frac{\sqrt{2}}{3}, \qquad (u \cdot v_{12}) = -\sqrt{\frac{2}{3}},$$

$$(q \cdot V) = \frac{1}{\sqrt{3}}, \qquad (q \cdot v_{11}) = \sqrt{\frac{2}{3}}, \qquad (q \cdot v_{12}) = \frac{\sqrt{2}}{3},$$

$$(q \cdot W) = \sqrt{3},$$

and

$$U = \left(\frac{3}{2}(S_1 - S_2), \frac{3}{4}(S_1 + S_2), \frac{3}{4}(S_1 + S_2)\right). \qquad (25)$$

When $a = b = c$ and $A = B = C$, the p-cages can easily be regular. The inner radius $r$ of a regular P-gon and its edge length $L$ satisfy

$$r_i = \frac{L}{2}\cotan\left(\frac{\pi}{P_i}\right) \qquad\qquad (26)$$

where $i$ refers to the type of face, and

$$r_1 = |U - V|, \qquad\qquad r_2 = |U - W|. \qquad\qquad (27)$$

so that

$$r_1^2 = \frac{3}{8}(S_1 - 3\,S_2)^2, \qquad\qquad r_2^2 = \frac{3}{8}(S_2 - 3\,S_1)^2. \qquad (28)$$

Equations (28) can easily be solved to obtain

$$S_1 = -\frac{1}{\sqrt{24}}(3\,r_2 + r_1), \qquad\qquad S_2 = -\frac{1}{\sqrt{24}}(3\,r_1 + r_2). \qquad (29)$$

We have to consider both positive and negative values of $r_1$ and $r_2$, leading to four solutions, but only keep positive values of $S_1$ and $S_2$ and the one for which they are both the largest.

Considering all the combinations of $P_1$ and $P_2$, we obtain the regular p-cages, with $a = b = c$ and $A = B = C$, listed in Table 1. We present all our results using six decimal paces as this corresponds to the accuracy generated by our computer program. For regular p-cages, the results are exact.

When $P_1 = P_2$, we can use a different approach. We consider a regular P-gon centred in the $x - y$ plane with vertices

$$\hat{n}_i = \rho\left(\sin\left(\frac{2\pi i}{P} + \phi\right), -\cos\left(\frac{2\pi i}{P} + \phi\right), 0\right)^t, \quad i = 0 \ldots P - 1. \qquad (30)$$

where $\rho = d/(2\sin(\pi/2P))$ is the radius of the circle containing the polygonal face of edge length $d$. The polygon is then rotated by an angle $\theta$ around the $x$ axis and then translated by a distance $-R$ along the $y$ axis, giving

$$\boldsymbol{n}_i = R_z(\sigma) \, R_x(\theta) \, \hat{\boldsymbol{n}}_i + (0, -R, 0)^t, \quad i = 0 \ldots P-1. \tag{31}$$

The vertices of the neighbour face are given by

$$\boldsymbol{m}_i = R_z\left(\frac{\pi}{2}\right) \boldsymbol{n}_i. \tag{32}$$

Defining

$$s_i = \frac{2i\pi}{P} + \phi, \tag{33}$$

we have

$$\boldsymbol{n}_i = \begin{pmatrix} \rho \sin s_i \cos \sigma - \rho \cos s_i \sin \sigma \cos \theta \\ -\rho \cos s_i \cos \sigma \cos \theta - \rho \sin s_i \sin \sigma - R \\ \rho \cos s_i \sin \theta \end{pmatrix},$$

$$\boldsymbol{m}_i = \begin{pmatrix} \rho \cos s_i \cos \sigma \cos \theta + \rho \sin s_i \sin \sigma + R \\ \rho \sin s_i \cos \sigma - \rho \cos s_i \sin \sigma \cos \theta \\ \rho \cos s_i \sin \theta \end{pmatrix}. \tag{34}$$

We must now impose some constraints on the vertices of the two faces so that they share one edge with their neighbours.

**Table 1.** List of all regular sp p-cages with $a = b = c = (P_1 - 3)/3$ and $A = B = C = (P_2 - 3)/3$.

| $P_1$ | $P_2$ | $S_1$ | $S_2$ | $S_2/S_1$ | $P_1$ | $P_2$ | $S_1$ | $S_2$ | $S_2/S_1$ |
|---|---|---|---|---|---|---|---|---|---|
| 6 | 6 | 0.707107 | 0.707107 | 1.000000 | 9 | 18 | 2.016882 | 1.420062 | 0.704088 |
| 6 | 9 | 1.018016 | 0.810743 | 0.796395 | 12 | 12 | 1.523603 | 1.523603 | 1.000000 |
| 6 | 12 | 1.319479 | 0.911231 | 0.690599 | 12 | 15 | 1.821394 | 1.622867 | 0.891003 |
| 6 | 15 | 1.617270 | 1.010494 | 0.624815 | 12 | 18 | 2.117369 | 1.721525 | 0.813049 |
| 9 | 9 | 1.121653 | 1.121653 | 1.000000 | 15 | 15 | 1.920657 | 1.920657 | 1.000000 |
| 9 | 12 | 1.423116 | 1.222141 | 0.858778 | 15 | 18 | 2.216633 | 2.019316 | 0.910983 |
| 9 | 15 | 1.720906 | 1.321404 | 0.767854 | 18 | 18 | 2.315291 | 2.315291 | 1.000000 |

### 7.1.2. Bottom of the Face

The bottom half of the p-cage must match the top half, so that, up to a $z$ translation (we ignore the $z$ component),

$$\boldsymbol{n}_0 = R_z\left(\frac{\pi}{2}\right) R_y(\pi) \, \boldsymbol{m}_{P-1}, \qquad \boldsymbol{n}_{P-1} = R_z\left(\frac{\pi}{2}\right) R_y(\pi) \, \boldsymbol{m}_0, \tag{35}$$

where

$$R_z\left(\frac{\pi}{2}\right) R_y(\pi) \, \boldsymbol{m}_i = \begin{pmatrix} -1 & 0 & 0 \\ 0 & 1 & 0 \\ 0 & 0 & -1 \end{pmatrix} \boldsymbol{n}_i, \tag{36}$$

which, after inserting the vectors $\boldsymbol{n}_i$ and $\boldsymbol{m}_i$ from (35) into (36), yields

$$(\sin s_0 + \sin s_{P-1}) \cos \sigma = (\cos s_{P-1} + \cos s_0) \sin \sigma \cos \theta$$
$$(\cos s_0 - \cos s_{P-1}) \cos \sigma \cos \theta = (\sin s_{P-1} - \sin s_0) \sin \sigma. \tag{37}$$

If $\sigma$ is nonzero, we can divide the first line of (37) by the second one, yielding

$$\tan(\sigma) = \frac{\sin s_0 + \sin s_{P-1}}{\cos s_0 + \cos s_{P-1}}\frac{1}{\cos\theta} = \frac{\cos s_0 - \cos s_{P-1}}{\sin s_{P-1} - \sin s_0}\cos\theta. \tag{38}$$

This implies that

$$\sin^2 s_{P-1} - \sin^2 s_0 = \cos^2\theta(\cos^2 s_0 - \cos^2 s_{P-1}) \tag{39}$$

and $\cos^2\theta = 1$. As a result, $\theta = 0$, which corresponds to a flat p-cage.

If $\sigma = 0$, then from (37), we have

$$\sin s_0 = -\sin s_{P-1}, \qquad \cos s_0 = \cos s_{P-1}. \tag{40}$$

Then, substituting (33) into the above, we obtain

$$\sin\phi = -\sin\left(\frac{2(P-1)\pi}{P} + \phi\right) = \sin\left(\frac{2\pi}{P} - \phi\right),$$
$$\cos\phi = \cos\left(\frac{2(P-1)\pi}{P} + \phi\right) = \cos\left(\frac{2\pi}{P} - \phi\right) \tag{41}$$

and this implies that

$$\phi = \frac{\pi}{P}. \tag{42}$$

7.1.3. Side Edges

There must also be indices $i_0$ and $j_0$ for which

$$\boldsymbol{n}_{i_0} = \boldsymbol{m}_{j_0+1}, \qquad\qquad \boldsymbol{n}_{i_0+1} = \boldsymbol{m}_{j_0}. \tag{43}$$

Using (34) and some algebra, we obtain

$$\frac{R}{\rho} = \sin s_{i_0} - \cos\theta\cos s_{i_0} = \sin s_{i_0+1} - \cos s_{j_0}\cos\theta \tag{44}$$

as well as

$$\cos\theta = \frac{\sin s_{i_0+1} - \sin s_{i_0}}{\cos s_{i_0+1} - \cos_{i_0}}. \tag{45}$$

For any value of $P$, we need to find the value for $i$ for which $\theta$ is in the range $[0, \pi/2]$ and for which $R/\rho$ is positive. This leads to the regular p-cages listed in [8] and given in Table 2.

**Table 2.** List of all regular sp p-cages with $P_1 = P_2$.

| Name | Name | Name |
|---|---|---|
| sp_P6_P6_1_1_1-1_1_1 | sp_P7_P7_1_1_2-1_1_2 | sp_P8_P8_1_2_2-1_2_2 |
| sp_P8_P8_1_1_3-1_1_3 | sp_P9_P9_2_2_2-2_2_2 | sp_P10_P10_2_2_3-2_2_3 |
| sp_P11_P11_2_3_3-2_3_3 | sp_P11_P11_2_2_4-2_2_4 | sp_P12_P12_3_3_3-3_3_3 |
| sp_P12_P12_2_2_5-2_2_5 | sp_P13_P13_3_3_4-3_3_4 | sp_P14_P14_3_3_5-3_3_5 |
| sp_P14_P14_3_4_4-3_4_4 | sp_P15_P15_4_4_4-4_4_4 | sp_P16_P16_4_4_5-4_4_5 |
| sp_P16_P16_3_5_5-3_5_5 | sp_P17_P17_4_5_5-4_5_5 | sp_P18_P18_5_5_5-5_5_5 |

### 7.2. Truncated Octahedron

The coordinates of the vertices of a truncated octahedron of edge length 1 are

$$(\pm\sqrt{2},\pm\tfrac{1}{2},0), \qquad (\pm\tfrac{1}{2},\pm\sqrt{2},0), \qquad (\pm\sqrt{2},0,\pm\tfrac{1}{2}),$$
$$(\pm\tfrac{1}{2},0,\pm\sqrt{2}), \qquad (0,\pm\sqrt{2},\pm\tfrac{1}{2},0), \qquad (0,\pm\tfrac{1}{2},\pm\sqrt{2},0). \tag{46}$$

The faces are mapped to each other via the following rotations:

$$\boldsymbol{\mathcal{V}}_{x2} = R_x(\pi)\boldsymbol{\mathcal{V}}_{x1}, \qquad\qquad \boldsymbol{\mathcal{V}}_{x3} = R_y(\pi)\boldsymbol{\mathcal{V}}_{x1}, \qquad\qquad \boldsymbol{\mathcal{V}}_{x4} = R_y(\pi)\boldsymbol{\mathcal{V}}_{x2},$$
$$\boldsymbol{\mathcal{V}}_{y1} = R_y\!\left(\tfrac{\pi}{2}\right)R_z\!\left(\tfrac{\pi}{2}\right)\boldsymbol{\mathcal{V}}_{x1}, \quad \boldsymbol{\mathcal{V}}_{y2} = R_y\!\left(\tfrac{\pi}{2}\right)R_z\!\left(\tfrac{\pi}{2}\right)\boldsymbol{\mathcal{V}}_{x2},$$
$$\boldsymbol{\mathcal{V}}_{y3} = R_z(\pi)\boldsymbol{\mathcal{V}}_{y1}, \qquad\qquad \boldsymbol{\mathcal{V}}_{y4} = R_z(\pi)\boldsymbol{\mathcal{V}}_{y2},$$
$$\boldsymbol{\mathcal{V}}_{z1} = R_z\!\left(\tfrac{\pi}{2}\right)R_y\!\left(\tfrac{\pi}{2}\right)\boldsymbol{\mathcal{V}}_{x1}, \quad \boldsymbol{\mathcal{V}}_{z2} = R_z\!\left(\tfrac{\pi}{2}\right)R_y\!\left(\tfrac{\pi}{2}\right)\boldsymbol{\mathcal{V}}_{x2},$$
$$\boldsymbol{\mathcal{V}}_{z3} = R_x(\pi)\boldsymbol{\mathcal{V}}_{z1}, \qquad\qquad \boldsymbol{\mathcal{V}}_{z4} = R_x(\pi)\boldsymbol{\mathcal{V}}_{z2}, \tag{47}$$

where $\boldsymbol{\mathcal{V}}$ stands for $\boldsymbol{V}$ or $\boldsymbol{W}$.

The normal to the face of the p-cage can be placed on the vertices of the truncated octahedron, but they can actually be rotated arbitrarily, so that the symmetries described above remaining valid.

We take the following (see Figure 5):

$$\boldsymbol{V}_{x1} = S(1, y_1, z_1)^t, \quad z_1 > 0,$$
$$\boldsymbol{W}_{x1} = S(1, y_2, z_2)^t, \quad y_1 > 0, \tag{48}$$

and the basis vectors for the reference faces are given by (15), where

$$\boldsymbol{v}_0 = \hat{e}_y, \qquad \boldsymbol{w}_0 = \hat{e}_y. \tag{49}$$

The bases for the other faces are obtained by applying the rotations relating these other faces to the reference faces, as described in (47).

We next use $\boldsymbol{V}, \boldsymbol{v}_{11}, \boldsymbol{v}_{12}, \boldsymbol{W}_i, \boldsymbol{w}_{i,1}$ and $\boldsymbol{w}_{i,2}$ for $i = 1, 2, 3$ in (18) and (16) to compute the vectors spanning the line of the intersection between adjacent faces $\boldsymbol{U}_i$ and $\boldsymbol{u}_i$. Then, the vertices on the face intersection are given by (22). The optimisation parameters are $V_1, W_1$, $t_1, t_2, t_3, T_4, t_5, t_6$, from Equation (22), as well as the coordinates, in the face plane, of the nonshared vertices.

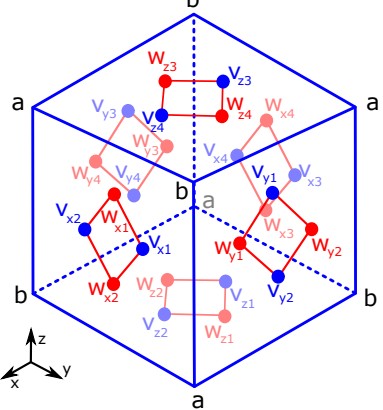

**Figure 5.** Labelling of the vertices of the reference faces of the truncated octahedron p-cage. The letter $v$ is used for the tyoe 1 faces while the letter $w$ is used for the type 2 faces. The labels $x, y, z$ are used for vertices located on the face normal to the corresponding axis.

Regular `Ato` P-Cages

Regular `Ato` p-cages can be obtained by tiling a square or a hexagon with some polygons (Figure 6).

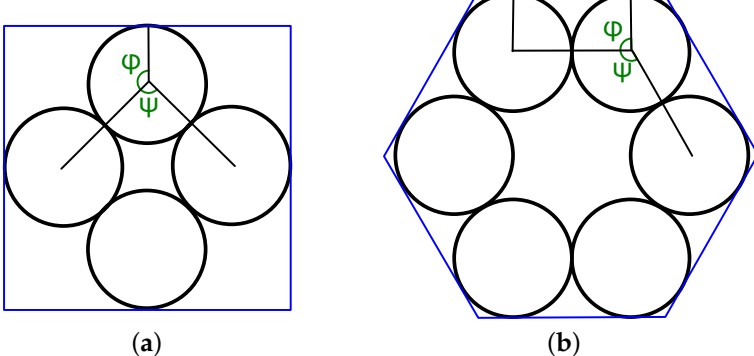

**Figure 6.** Tiling of a square or a hexagon using polygons. (**a**) square: $\varphi = 3\pi/4$, $\psi = 2\pi/4$, (**b**) hexagon: $\varphi = 2\pi/4$, $\psi = 2\pi/3$,

When the face contributes $q$ edges to a hole, the rotation is $2\pi(q+1)/P$. We must then impose that $\varphi = 2\pi(q_2+1)/P$ and $\psi = 2\pi(q_1+1)/P$. For the square, we have $q_2 + 1 = 3P/8$ and $q_1 + 1 = P/4$, implying that $P$ is a multiple of 8 and the regular p-cages of that type are `Ato_P8_P8_1_2_2-1_2_2` and `Ato_P16_P16_3_5_5-3_5_5`. For the hexagon, $q_2 + 1 = P/4$ and $q_1 + 1 = P/3$, so $P$ must be a multiple of both 4 and 3, and the regular p-cage of that type is `Ato_P12_P12_2_3_4-2_3_4` and the identical p-cage is `Ato_P12_P12_2_4_3-2_4_3`.

If we join together the vertices shared by adjacent faces, we obtain an irregular hexagon that we call the subface (see Figure 7a). To obtain a regular `Ato` p-cage, the face must form holes centred on the vertices of the underlying octahedron (see Figure 7b,c). Faces *I* and *II* contribute, respectively, $q_1$ edges and $Q_1$ edges to that hole.

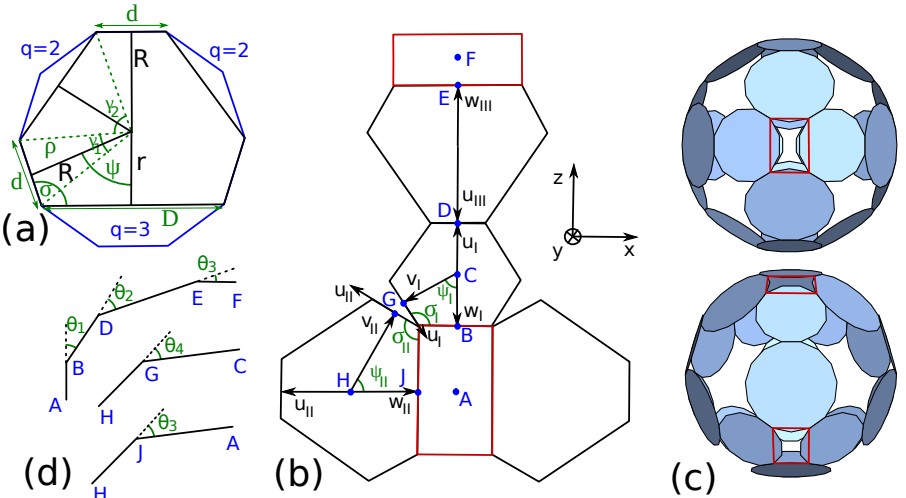

**Figure 7.** Regular Ato p-cage: (**a**) p-cage irregular hexagon subface of the p-cage. (**b**) Planar unfolding of the p-cage. The red rectangle are centred on the vertices of the underlying octagon. (**c**) Regular cages `Ato_P12_P15_3_3_3-2_5_5`. (**d**) Side view joining two vertices of the underlying octahedron.

If $\rho$ is the radius of the circle containing the regular polygon (see Figure 7a), $R$ denotes the distance between the centre of the face and the centre of a shared edge. $r$ is used to denote the distance between the centre of the face and the centre of the large edge of the subface hexagon. We use $\gamma_q = 2\pi q/P$ to denote the angle spanned by $q$ segments, $d$ is the

length of a face edge, and $D$ the length of the edge of the subface hexagon. The angle $\psi$ between the face edge and the adjacent subface edge is $\pi(q+1)/P$. We then have

$$R = \frac{d}{2 \tan \frac{\pi}{P}}, \qquad \rho = \frac{d}{2 \sin \frac{\pi}{P}}, \qquad r = \rho \cos \left( \frac{\gamma_q}{2} \right) = \frac{d}{2} \frac{\cos \left( \frac{\pi q}{P} \right)}{\sin \left( \frac{\pi}{P} \right)}. \tag{50}$$

Seen from the side, the faces I and III form, respectively, an angle $\alpha_1$ and $\alpha_1 + \alpha_2$ with the $x - z$ plane and face III forms an angle $\alpha_3$ with the $x - y$ plane (Figure 7d). Note also that, by symmetry, the angle $H - J - A$ in Figure 7d is the same as the angle $D - E - F$: $\alpha_3$.

Using lowercase letters for the vectors in the unfolded diagram and uppercase letters for the vectors in the p-cage, we have

$$
\begin{aligned}
\boldsymbol{v}_I &= R_I(-\sin \psi_I, 0, -\cos \psi_I), & \boldsymbol{v}_{II} &= R_{II}(\cos \psi_{II}, 0, \sin \psi_{II}), \\
\boldsymbol{w}_I &= r_I(0, 0, -1), & \boldsymbol{w}_{II} &= r_{II}(1, 0, 0), \\
\boldsymbol{u}_I &= R_I(0, 0, 1), & \boldsymbol{u}_{II} &= R_{II}(-1, 0, 0), \\
\boldsymbol{n}_I &= (\cos \psi_I, 0, -\sin \psi_I), & \boldsymbol{n}_{II} &= (-\sin \psi_{II}, 0, \cos \psi_{II}).
\end{aligned} \tag{51}
$$

The vectors in face $I$ are rotated by an angle $\theta_1$ around the $x$ axis, and face $II$ is rotated an angle $\theta_3$ around the $z$ axis. The matrices are

$$R_x(\theta_1) = \begin{pmatrix} 1 & 0 & 0 \\ 0 & \cos \theta_1 & \sin \theta_1 \\ 0 & -\sin \theta_1 & \cos \theta_1 \end{pmatrix}, \qquad R_z(\theta_3) = \begin{pmatrix} \cos \theta_3 & \sin \theta_3 & 0 \\ -\sin \theta_3 & \cos \theta_3 & 0 \\ 0 & 0 & 1 \end{pmatrix}. \tag{52}$$

We multiply the vectors with index $I$ defined in (51) by $R_x(\theta_1)$, while the vectors with index $II$ are multiplied by $R_z(\theta_3)$. The vectors obtained are denoted by the corresponding uppercase letter:

$$
\begin{aligned}
\boldsymbol{V}_I &= R_x(\theta_1)\boldsymbol{v}_I = R_I(-\sin \psi_I, -\cos \psi_I \sin \theta_1, -\cos \psi_I \cos \theta_1), \\
\boldsymbol{V}_{II} &= R_z(\theta_3)\boldsymbol{v}_{II} = R_{II}(\cos \psi_{II} \cos \theta_3, \cos \psi_{II} \sin \theta_3, \sin \psi_{II}), \\
\boldsymbol{W}_I &= R_x(\theta_1)\boldsymbol{w}_I = r_I(0, -\sin \theta_1, -\cos \theta_1), \\
\boldsymbol{W}_{II} &= R_z(\theta_3)\boldsymbol{w}_{II} = r_{II}(\cos \theta_3, \sin \theta_3, 0), \\
\boldsymbol{U}_I &= R_x(\theta_1)\boldsymbol{u}_I = R_I(0, \sin \theta_1, \cos \theta_1), \\
\boldsymbol{U}_{II} &= R_z(\theta_3)\boldsymbol{u}_{II} = R_{II}(-\cos \theta_3, -\sin \theta_3, 0), \\
\boldsymbol{N}_I &= R_x(\theta_1)\boldsymbol{n}_I = (\cos \psi_I, -\sin \psi_I \sin \theta_1, -\sin \psi_I \cos \theta_1), \\
\boldsymbol{N}_{II} &= R_z(\theta_3)\boldsymbol{n}_{II} = (-\sin \psi_{II} \cos \theta_3, -\sin \psi_{II} \sin \theta_3, \cos \psi_{II}).
\end{aligned} \tag{53}
$$

After folding the planar faces into the p-cage, the following conditions must be set so that the faces share 1 edge:

$$(\boldsymbol{N}_I \cdot \boldsymbol{V}_{II}) = 0, \qquad (\boldsymbol{N}_{II} \cdot \boldsymbol{V}_I) = 0 \tag{54}$$

which, after inserting the vectors defined in (53), yields

$$0 = \cos \psi_I \cos \psi_{II} \cos \theta_3 - \sin \psi_I \sin \theta_1 \cos \psi_{II} \sin \theta_3 - \sin \psi_I \cos \theta_1 \sin \psi_{II} \tag{55}$$

$$0 = \sin \psi_{II} \cos \theta_3 \sin \psi_I - \sin \psi_{II} \sin \theta_3 \cos \psi_I \sin \theta_1 - \cos \psi_{II} \cos \psi_I \cos \theta_1. \tag{56}$$

Multiplying (55) by $\sin \psi_{II} \cos \psi_I$ and (56) by $\sin \psi_I \cos \psi_{II}$ and adding the two together, we obtain

$$\cos \theta_3 = \frac{\sin(2\psi_I)}{\sin(2\psi_{II})} \cos \theta_1. \tag{57}$$

Substituting (57) into (55), we obtain, after some manipulations,

$$\cos^2 \theta_1 = \frac{(1 \pm \cos(2\psi_I))(1 \pm \cos(2\psi_{II}))}{\sin^2(2\psi_I)}. \tag{58}$$

In (58), the $\pm$ signs must be identical, and $\cos(\theta_1)$ must be positive, so we have two solutions to consider.

We must impose other constraints. First of all, to ensure the convexity of the p-cage, we must have $\sigma_I + \sigma_{II} \leq 3\pi/2$ where $\sigma_i = \pi - \psi_i$. Then, the centres of the holes must be located at the vertices of an octahedron, or, in other words, the segment $A - F$ must form a 45-degree angle with the vertical axis. With

$$D_I = r_I \tan(\frac{\pi q_1}{P_1}), \qquad D_{II} = r_{II} \tan(\frac{\pi Q_1}{P_2}), \tag{59}$$

the coordinates of these two points are

$$A_y = -(r_I + R_I)\sin(\theta_1) - (r_{II} + R_{II})\sin(\theta_1 + \theta_2) - \frac{D_I}{2}, \tag{60}$$

$$A_x = A_z = 0, \tag{61}$$

$$F_z = (r_I + R_I)\cos(\theta_1) + (r_{II} + R_{II})\cos(\theta_1 + \theta_2) + \frac{D_{II}}{2}, \tag{62}$$

$$A_x = A_z = 0. \tag{63}$$

The constraint $F_Z = -A_y$ must also be satisfied, which is, more explicitly:

$$\Delta_O = (r_I + R_I)(\sin(\theta_1) - \cos(\theta_1)) + (r_{II} + R_{II})(\sin(\theta_1 + \theta_2) - \cos(\theta_1 + \theta_2)) + \frac{D_I - D_{II}}{2}$$
$$= 0. \tag{64}$$

Finally, we must impose the constraint that the faces are not crossing each other. This means that for $q_1$ even $(\rho_1 - r_1)\cos\theta_1 \leq D_2/2$ and, for $q_1$ odd, $(R_1 - r_1)\cos\theta_1 \leq D_2/2$. For $Q_1$, even $\rho_2 - r_2\cos\theta_3 \leq D_1/2$ must hold while for $Q_1$, odd $(R_2 - r_2)\cos\theta_3 \leq D_1/2$.

The list of regular `Ato` p-cages is given in Table 3.

**Table 3.** Regular `Ato` p-cages.

| Name | Name | Name |
|---|---|---|
| `Ato_P6_P10_1_1_1-1_3_3` | `Ato_P10_P12_1_3_3-3_3_3` | `Ato_P15_P15_2_5_5-4_4_4` |
| `Ato_P6_P15_1_1_1-2_5_5` | `Ato_P11_P11_2_3_3-2_3_3` | `Ato_P15_P18_2_5_5-5_5_5` |
| `Ato_P8_P8_1_2_2-1_2_2` | `Ato_P12_P15_3_3_3-2_5_5` | `Ato_P16_P16_3_5_5-3_5_5` |
| `Ato_P9_P10_2_2_2-1_3_3` | `Ato_P12_P12_2_3_4-2_3_4` | `Ato_P17_P17_4_5_5-4_5_5` |
| `Ato_P9_P15_2_2_2-2_5_5` | `Ato_P14_P14_3_4_4-3_4_4` | |

Note that `Ato_P13_P17_2_4_4-4_5_5` is nearly regular with a deformation $\Delta_l = \Delta_a = 6.66 \cdot 10^{-5}$, and the reason for this is that $\Delta_O = 0.00169$.

### 7.3. Truncated Cuboctahedron

We inscribe a truncated cuboctahedron in a cube of edge length 4 so that the octagons are contained inside the face of the cube. We label the vertices as $\mathcal{V}_{\pm\sigma i}$, where $\sigma$ stands for

$x$, $y$, or $z$, and corresponds to the octagon contained in the plane $\sigma = \pm 2$. $i$ is the index of the octagon vertices, and they have the following coordinates (see Figure 8):

$$\boldsymbol{\mathcal{V}}_{x1} = (2, \sqrt{2}, 2 - \sqrt{2})^t, \qquad\qquad \boldsymbol{\mathcal{V}}_{x2} = R_x(\tfrac{\pi}{4})\boldsymbol{\mathcal{V}}_{x1} = (2, 2 - \sqrt{2}, \sqrt{2})^t,$$

$$\boldsymbol{\mathcal{V}}_{x3} = R_x(\tfrac{\pi}{2})\boldsymbol{\mathcal{V}}_{x1} = (2, \sqrt{2} - 2, \sqrt{2})^t, \qquad \boldsymbol{\mathcal{V}}_{x4} = R_x(\tfrac{\pi}{2})\boldsymbol{\mathcal{V}}_{x2} = (2, -\sqrt{2}, 2 - \sqrt{2})^t,$$

$$\boldsymbol{\mathcal{V}}_{x5} = R_x(\pi)\boldsymbol{\mathcal{V}}_{x1} = (2, -\sqrt{2}, \sqrt{2} - 2)^t, \qquad \boldsymbol{\mathcal{V}}_{x6} = R_x(\pi)\boldsymbol{\mathcal{V}}_{x2} = (2, \sqrt{2} - 2, -\sqrt{2})^t,$$

$$\boldsymbol{\mathcal{V}}_{x7} = R_x(-\tfrac{\pi}{2})\boldsymbol{\mathcal{V}}_{x1} = (2, 2 - \sqrt{2}, -\sqrt{2})^t, \quad \boldsymbol{\mathcal{V}}_{x8} = R_x(-\tfrac{\pi}{2})\boldsymbol{\mathcal{V}}_{x2} = (2, \sqrt{2}, \sqrt{2} - 2)^t. \quad (65)$$

The vertices are then obtained using the following rotations:

$$\boldsymbol{\mathcal{V}}_{yi} = R_z(\tfrac{\pi}{2})\boldsymbol{\mathcal{V}}_{xi}, \qquad\qquad \boldsymbol{\mathcal{V}}_{zi} = R_y(-\tfrac{\pi}{2})\boldsymbol{\mathcal{V}}_{xi},$$

$$\boldsymbol{\mathcal{V}}_{-xi} = R_z(\pi)\boldsymbol{\mathcal{V}}_{xi}, \qquad\qquad \boldsymbol{\mathcal{V}}_{-yi} = R_z(-\tfrac{\pi}{2})\boldsymbol{\mathcal{V}}_{xi}, \qquad\qquad \boldsymbol{\mathcal{V}}_{-zi} = R_y(\tfrac{\pi}{2})\boldsymbol{\mathcal{V}}_{xi}. \quad (66)$$

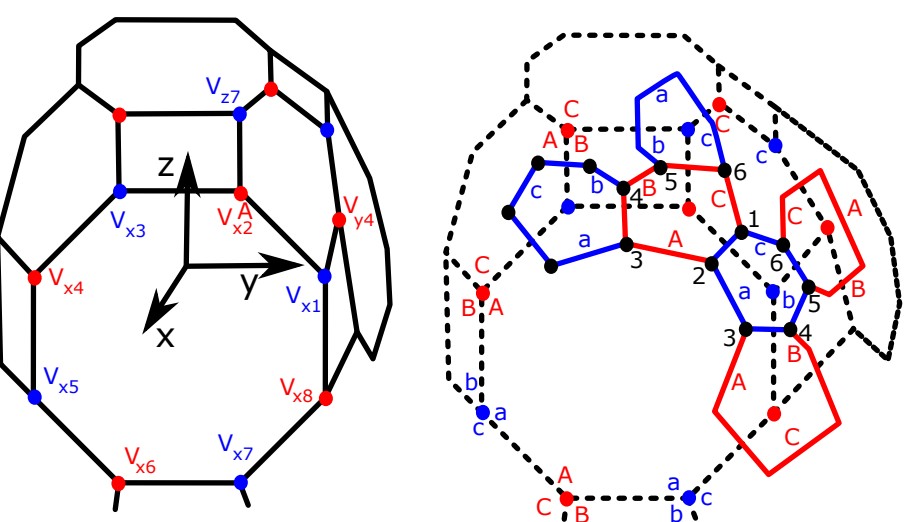

**Figure 8.** Labelling of the vertices of the reference faces of the truncated cuboctahedron p-cage. The letter $v$ is used for the type 1 faces while the letter $w$ is used for the type 2 faces.

We then place the reference faces on the following vertices:

$$V = \boldsymbol{\mathcal{V}}_{x1}, \qquad W_1 = \boldsymbol{\mathcal{V}}_{x2}, \qquad W_2 = \boldsymbol{\mathcal{V}}_{x8}, \qquad W_3 = \boldsymbol{\mathcal{V}}_{y4}, \qquad (67)$$

and take

$$v_0 = \hat{e}_y, \qquad w_0 = \hat{e}_y. \qquad (68)$$

We then use $V, v_{11}, v_{12}, W_i, w_{i,1}$, and $w_{i,2}$ for $i = 1, 2, 3$ in (18) and (16) to compute the vectors spanning the line of intersection between adjacent faces $U_i$ and $u_i$. The vertices on the face intersection are given by (22). The optimisation parameters are $V_1$, $W_1$, $t_1, t_2, t_3, T_4, t_5, t_6$, from Equation (22), as well as the coordinates, in the face plane, of the nonshared vertices.

Regular `Atco` P-Cages

To obtain a regular `Atco` p-cage, we must tile a hexagon with six polyhedra, which, as seen above, implies that $P$ must be a multiple of 3 and 4. So, the only regular `Atco` p-cage, with $P \le 18$, is the `Atco_P12_P12_4_2_3-4_2_3`. It is obtained by tiling each face of an octahedron with six regular dodecagons.

### 7.4. Truncated Icosidodecahedron

The truncated icosidodecahedron is made of 12 decagons, 20 hexagons and 30 squares. The decagons are centred on the vertices of an icosahedron that have the following coordinates (see Figure 9):

$$\mathcal{V}_{x,i} = (0, \pm 1, \pm \phi_g)^t, \qquad \mathcal{V}_{y,i} = (\pm \phi_g, 0, \pm 1)^t, \qquad \mathcal{V}_{z,i} = (\pm 1, \pm \phi_g, 0)^t, \tag{69}$$

where $\phi_g = (1 + \sqrt{5})/2$ is the golden ratio.

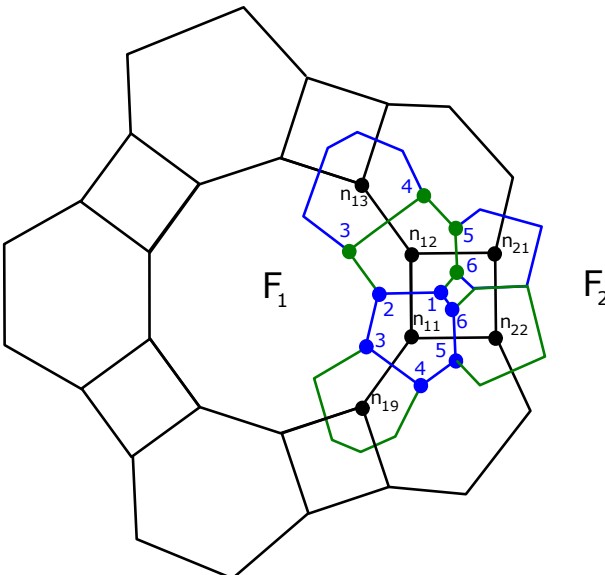

**Figure 9.** Labelling of the vertex coordinates for the p-cages derived from the truncated icosidodecahedron. The blue edges and node numbers correspond to the faces of type 1 while the green edges and nudes correspond to type 2 faces. The nodes are ordered respectively anti-clockwise and clockwise for the type 1 and type 2 faces.

The angle $2\theta$ between two decagons of the truncated icosidodecahedron is the angle between two adjacent vertices of the icosahedron. Taking $v = (0, 1, \phi_g,)^t$ and $w = (0, -1, \phi_g)^t$, we have $\cos(2\theta) = (\phi_g^2 - 1)/(\phi_g^2 + 1) = (1 + \sqrt{5})/(5 + \sqrt{5})$. Then, $\sin(\theta) = \sqrt{(1 - \cos(2\theta))/2} = \sqrt{2/(5 + \sqrt{5})}$. $\cos(\theta) = \sqrt{(3 + \sqrt{5})/(5 + \sqrt{5})}$.

If $R$ denotes the distance between the centre of the truncated icosidodecahedron and the centre of a decagonal face, and $r$ denotes the inner radius of the decagonal face (see Figure 10), if we look at the cross-section of the truncated icosidodecahedron going through the centres of two adjacent decagons and cutting a square face in two, we have

$$r = \frac{L}{2}\cotan(\frac{\pi}{10}), \qquad\qquad \delta r = \frac{L}{2\sin(\beta)},$$
$$R = (r + \delta r)\cotan(\theta), \qquad\qquad \beta = \frac{\pi}{2} - \theta. \tag{70}$$

Then

$$r = \frac{L}{2}\cotan\left(\frac{\pi}{10}\right) = \frac{L}{2}\left(\sqrt{5 + 2\sqrt{5}}\right) \qquad\qquad ,$$
$$R = \frac{L}{2}\left(\cotan\left(\frac{\pi}{10}\right) + \frac{1}{\cos(\theta)}\right) = \frac{L}{2}\left(\sqrt{5 + 2\sqrt{5}} + \sqrt{\frac{5 + \sqrt{5}}{3 + \sqrt{5}}}\right). \tag{71}$$

For $L = 1$, we have $r \approx 1.538841$ and $R \approx 2.126627$.

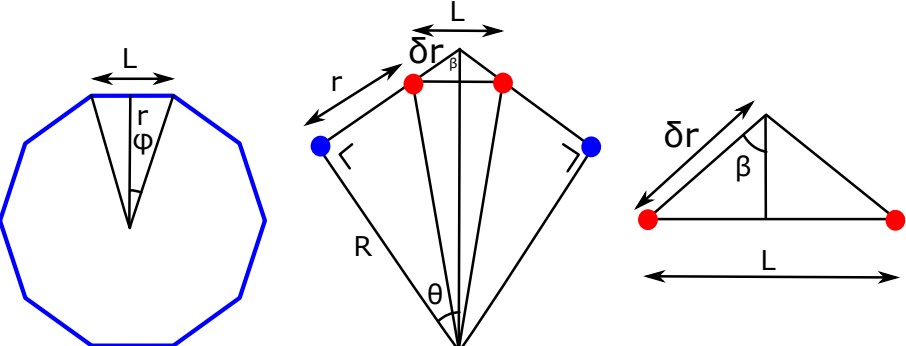

**Figure 10.** Derivation of the vertex coordinates for the truncated icosidodecahedron.

We place the centre of the first octagon at

$$F_1 = (R, 0, 0)^t \tag{72}$$

and the 10 vertices of that octagon at

$$n_{11} = \left(R, r, -r\tan(\frac{\pi}{10})\right) = \left(R, r, -r\sqrt{1 - \frac{2}{\sqrt{5}}}\right)^t = \left(R, r, -\frac{L}{2}\right)^t,$$

$$n_{1i} = R_x\left(i\frac{\pi}{5}\right)v_1 \quad i = 2\ldots 9. \tag{73}$$

Defining

$$g = (R\cos(\theta), R\sin(\theta), 0)^t = \left(R\sqrt{\frac{3+\sqrt{5}}{5+\sqrt{5}}}, R\sqrt{\frac{2}{5+\sqrt{5}}}, 0\right)^t, \tag{74}$$

the five decagons surrounding the first one are then given by

$$F_{2+i} = R_x\left(i\frac{2\pi}{5}\right)R_g(\pi)F_1, \quad i = 0\ldots 4, \tag{75}$$

while the remaining six decagons are

$$F_{7+i} = R_z(\pi)F_{1+i}, \quad i = 0\ldots 5. \tag{76}$$

The vertices $n_{ji}$ around the decagon $j$ are the obtained by applying the rotation in (75) to the vertices (73). The reference faces are then placed on the vertices

$$V = n_{11}, \quad W_1 = n_{1,2}, \quad W_2 = R_x(\frac{-\pi}{5})W_1 = n_{19}, \quad W_3 = R_g(\pi)W_1 = n_{22}. \tag{77}$$

We now take

$$v_0 = \hat{e}_y, \qquad\qquad w_0 = \hat{e}_y. \tag{78}$$

Next, we use $V, v_{11}, v_{12}, W_i, w_{i,1}$ and $w_{i,2}$ for $i = 1, 2, 3$ in (18) and (16) to compute the vectors spanning the line of the intersection between adjacent faces $U_i$ and $u_i$. The vertices on the face intersection are given by (22), and the optimisation parameters are $V_1$, $W_1, t_1, t_2, t_3, T_4, t_5, t_6$, from Equation (22), as well as the coordinates, in the face plane, of the nonshared vertices.

There are no convex regular `Atid` p-cages.

### 7.5. Rhombic Dodecahedron

The rhombic dodecahedron is the dual of the cuboctahedron, and as both types of faces must contribute the same number of edges to every hole, by symmetry, the faces must be spanned by the faces of the cuboctahedron.

A cuboctahedron of edge length $\sqrt{2}$ has vertices at the coordinates $(\pm 1, \pm 1, 0)$, $(\pm 1, 0, \pm 1)$ and $(0, \pm 1, \pm 1)$. Hence, we can take

$$V_1 = (1,0,0)^t, \qquad\qquad W_1 = \frac{2}{3}(1,1,1)^t \qquad (79)$$

and

$$
\begin{aligned}
v_{11} &= (0,1,0)^t, &\qquad v_{12} &= (0,0,1)^t, \\
w_{11} &= \frac{1}{\sqrt{2}}(-1,1,0)^t, &\qquad w_{12} &= \frac{1}{\sqrt{6}}(1,1,-2)^t.
\end{aligned} \qquad (80)
$$

We then have

$$
\begin{aligned}
V_2 &= R_z(\tfrac{\pi}{2})V_1, & V_3 &= R_z(\pi)V_1, & V_4 &= R_z(-\tfrac{\pi}{2})V_1, & V_5 &= R_y(-\tfrac{\pi}{2}), V_1 \\
V_6 &= R_y(\tfrac{\pi}{2})V_1, & W_2 &= R_z(\tfrac{\pi}{2})W_1, & W_3 &= R_z(\pi)W_1, & W_4 &= R_z(-\tfrac{\pi}{2})V_1, \\
W_5 &= R_z(\tfrac{\pi}{2})W_1, & W_6 &= R_z(\pi)W_1, & W_7 &= R_y(\pi)V_1, & W_8 &= R_z(\pi)W_5.
\end{aligned} \qquad (81)
$$

Using $g = (1,1,1)/\sqrt{3}$, we also have the following relationships between the vertices (see Figure 11):

$$
\begin{aligned}
n_3 &= R_x(\tfrac{\pi}{2})n_1, & n_4 &= R_x(\tfrac{\pi}{2})n_2, & n_5 &= R_x(\pi)n_1, & n_6 &= R_x(\pi)n_2, \\
n_7 &= R_x(\tfrac{-\pi}{2})n_1, & n_8 &= R_x(-\tfrac{\pi}{2})n_2, & m_3 &= R_g(-\tfrac{2\pi}{3})n_1, \\
m_4 &= R_g(-\tfrac{2\pi}{3})n_2, & m_5 &= R_g(\tfrac{2\pi}{3})n_1, & m_6 &= R_g(\tfrac{2\pi}{3})n_2.
\end{aligned} \qquad (82)
$$

As all $q_i$ are identical and so are $Q_i$, the p-cages derived from this graph are all regular. Indeed, the inner radius $r$ of a regular P-gon and its edge length $L$ satisfy (26) and (27).

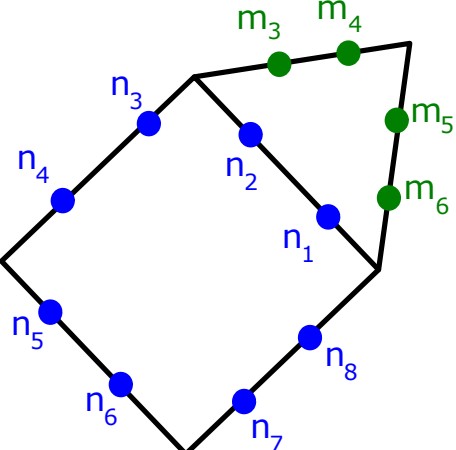

**Figure 11.** Labelling of the vertices of the reference faces of the rhombic dodecahedron p-cage. The vertices of type 1 faces are denoted $n$ and numbered anti-clockwise while the vertices of type 2 faces are denoted $m$ and numbered clockwise.

We then have

$$
\boldsymbol{V} = (S_2, 0, 0), \qquad\qquad \boldsymbol{W} = \left(\frac{2}{3}S_1, \frac{2}{3}S_1, \frac{2}{3}S_1\right),
$$

$$
\boldsymbol{v}_1 = (0, 1, 0), \qquad\qquad \boldsymbol{w}_1 = (-1, 1, 0)/\sqrt{2},
$$

$$
\boldsymbol{v}_2 = (0, 0, 1), \qquad\qquad \boldsymbol{w}_1 = (1, 1, -2)/\sqrt{6},
$$

$$
\boldsymbol{p} = \boldsymbol{v}_1 \times \boldsymbol{v}_2 = (1, 0, 0), \qquad\qquad \boldsymbol{q} = \boldsymbol{w}_1 \times \boldsymbol{w}_2 = (-1, -1, -1)/\sqrt{3},
$$

$$
\boldsymbol{u} = \boldsymbol{q} \times \boldsymbol{p} = (0, -1, 1)/\sqrt{3}. \tag{83}
$$

From (18), we have

$$
\boldsymbol{U} = \left(S_2, \frac{2S_1 - S_2}{2}, \frac{2S_1 - S_2}{2}\right). \tag{84}
$$

and, as a result,

$$
r_2 = |\boldsymbol{U} - \boldsymbol{V}| = \left|\frac{2S_1 - S_2}{\sqrt{2}}\right|,
$$

$$
r_1 = |\boldsymbol{U} - \boldsymbol{W}| = \left|\frac{3S_2 - 2s_1}{\sqrt{6}}\right|. \tag{85}
$$

This can easily be solved to obtain

$$
S_1 = \frac{1}{2}\left(\frac{3}{\sqrt{2}}r_2 + \sqrt{\frac{3}{2}}r_1\right),
$$

$$
S_2 = \left(\sqrt{\frac{3}{2}}r_1 + \frac{1}{\sqrt{2}}r_2\right). \tag{86}
$$

This leads to the regular p-cages listed in Table 4.

**Table 4.** List of all regular DArd p-cages with $a = b = c = (P_1 - 3)/3$ and $A = B = C = D = (P_2 - 4)/4$.

| $P_1$ | $P_2$ | $S_1$ | $S_2$ | $S_2/S_1$ | $P_1$ | $P_2$ | $S_1$ | $S_2$ | $S_2/S_1$ |
|---|---|---|---|---|---|---|---|---|---|
| 6 | 8 | 1.810660 | 1.914214 | 1.057191 | 12 | 12 | 3.121921 | 3.604884 | 1.154701 |
| 6 | 12 | 2.509549 | 2.380139 | 0.948433 | 12 | 16 | 3.808852 | 4.062838 | 1.066683 |
| 6 | 16 | 3.196479 | 2.838093 | 0.887881 | 15 | 12 | 3.419712 | 4.200465 | 1.228310 |
| 9 | 8 | 2.121570 | 2.536033 | 1.195357 | 15 | 16 | 4.106642 | 4.658419 | 1.134362 |
| 9 | 12 | 2.820459 | 3.001959 | 1.064351 | 18 | 12 | 3.715687 | 4.792416 | 1.289779 |
| 9 | 16 | 3.507389 | 3.459912 | 0.986464 | 18 | 16 | 4.402618 | 5.250370 | 1.192556 |
| 12 | 8 | 2.423033 | 3.138958 | 1.295467 | 21 | 16 | 4.697562 | 5.840259 | 1.243253 |

### 7.6. Rhombic Triacontahedron

The faces of the p-cages are inscribed inside the faces of the icosidodecahedron, and the center of pentagonal faces are given by (69) (see Figure 12). There are regular p-cages of this type. Indeed, the coordinates of the centres of the triangular faces are

$$
\boldsymbol{W} = \frac{1}{3}\big((0, 1, \phi_g) + (0, -1, \phi_g) + (1, \phi_g, 0)\big) = \frac{1}{3}(1, \phi_g, 2\phi_g). \tag{87}
$$

The angle between $\boldsymbol{v}_1 = (0, 1, \phi_g)$ and $\boldsymbol{w}_1 = (1, \phi_g, 2\phi_g)$ is such that

$$
\cos(\theta) = \frac{(\boldsymbol{v}_1 \cdot \boldsymbol{w}_1)}{|\boldsymbol{v}_1||\boldsymbol{w}_1|} = \frac{7 + 3\sqrt{5}}{\sqrt{110 + 42\sqrt{5}}}. \tag{88}
$$

The coordinates of the vertices of the icosidodecahedron are

$$q_{z1} = (\phi_g, 0, 0), \qquad V_{z2} = (-\phi_g, 0, 0), \qquad q_{y1} = (\phi_g, 0, 0),$$
$$V_{y2} = (-\phi_g, 0, 0), \qquad q_{z1} = (0, 0, \phi_g), \qquad V_{z2} = (0, 0, -\phi_g),$$
$$q_{x\pm\pm\pm} = (\pm 1, \pm\phi_g^2, \pm\phi_g), \quad q_{y\pm\pm\pm} = (\pm\phi_g, \pm 1, \pm\phi_g^2), \quad q_{z\pm\pm\pm} = (\pm\phi_g^2, \phi_g, \pm 1). \tag{89}$$

Then, the coordinates of the reference pentagon are (see Figure 12)

$$g_1 = \left(\frac{1}{2}, \frac{\phi_g}{2}, \frac{\phi_g^2}{2}\right), \qquad g_2 = (0, 0, \phi_g), \qquad g_3 = \left(\frac{1}{2}, -\frac{\phi_g}{2}, \frac{\phi_g^2}{2}\right),$$
$$g_4 = \left(\frac{\phi_g^2}{2}, -\frac{1}{2}, \frac{\phi_g}{2}\right), \qquad g_5 = \left(\frac{\phi_g^2}{2}, \frac{1}{2}, \frac{\phi_g}{2}\right), \tag{90}$$

and the face is centred at

$$g_p = \left(\frac{1 + \phi_g^2}{5}, 0, \frac{2\phi_g + \phi_g^2}{5}\right). \tag{91}$$

The coordinates of the reference triangle are

$$g_1 = \left(\frac{1}{2}, \frac{\phi_g}{2}, \frac{\phi_g^2}{2}\right), \qquad g_6 = \left(-\frac{1}{2}, \frac{\phi_g}{2}, \frac{\phi_g^2}{2}\right), \qquad g_7 = (0, 0, \phi_g), \tag{92}$$

and the face is centred at

$$g_t = \left(0, \frac{\phi_g}{3}, \frac{\phi_g + \phi_g^2}{3}\right). \tag{93}$$

We take

$$V = g_p, \qquad W = g_t, \qquad v_0 = (0, 1, 0), \qquad w_0 = (-1, 0, 0). \tag{94}$$

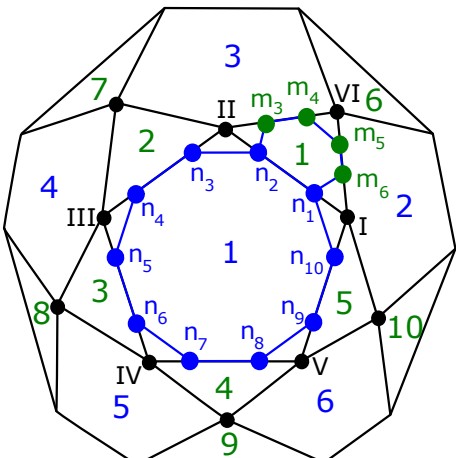

**Figure 12.** Labelling of the vertices of the reference faces of the rhombic triacontahedron p-cage as mapping the faces on an icosidodecahedron. Vertices *I*, *I*, *II*, *IV*, *V*, and *VI* are, respectively, $g_1$, $g_2$, $g_3$, $g_4$, $g_5$ and $g_6$. The vertices of type 1 faces are denoted $n$ and numbered anti-clockwise while the vertices of type 2 faces are denoted $m$ and numbered clockwise.

As all the $q_i$ are identical, all the p-cages are regular.

Regular `DArt` P-Cages

As all $q_i$ are identical and so are $Q_i$, the p-cages derived from this graph are regular, as we now prove. Taking $v_1 = v_0$ and $w_1 = w_0$, from (94), we have

$$V = S_2 \left( \frac{1 + \phi_g^2}{5}, 0, \frac{2\phi_g + \phi_g^2}{5} \right), \qquad W = S_1 \left( 0, \frac{\phi_g}{3}, \frac{\phi_g + \phi_g^2}{3} \right),$$

$$v_1 = (0, 1, 0), \qquad\qquad\qquad w_1 = (-1, 0, 0). \tag{95}$$

The vector $v_2$, which is perpendicular to both $v_1$ and $V$, and, similarly, $w_2$, which is perpendicular to both $w_1$ and $W$, are given by

$$v_2 = \left( -\frac{3\sqrt{5} + 5}{4\sqrt{5} + 10}, 0, \frac{\sqrt{5} + 5}{4\sqrt{5} + 10} \right), \qquad w_2 = \left( -\frac{4 + 2\sqrt{5}}{7 + 3\sqrt{5}}, 0, \frac{1 + \sqrt{5}}{7 + 3\sqrt{5}} \right). \tag{96}$$

From (16), we have

$$p = v_1 \times v_2 = \left( \frac{\sqrt{5} + 5}{4\sqrt{5} + 10}, 0, \frac{3\sqrt{5} + 5}{4\sqrt{5} + 10} \right),$$

$$q = w_1 \times w_2 = \left( 0, \frac{1 + \sqrt{5}}{7 + 3\sqrt{5}}, \frac{4 + 2\sqrt{5}}{7 + 3\sqrt{5}} \right), \tag{97}$$

and

$$u = q \times p = \left( \frac{4\sqrt{5} + 10}{29\sqrt{5} + 65}, \frac{7\sqrt{5} + 15}{29\sqrt{5} + 65}, -\frac{3\sqrt{5} + 5}{29\sqrt{5} + 65} \right). \tag{98}$$

From (18), we obtain a very complicated analytical expression for $U$ that can be approximated numerically as

$$\begin{aligned} U \approx (\ & 1.963525491562422 \, S_2 - 1.713525491562421 \, S_1, \\ & - 1.059016994374947 \, S_2 + 1.463525491562421 \, S_1, \\ & 0.4045084971874737 \, S_2 + 1.059016994374948 \, S_1 ). \end{aligned} \tag{99}$$

In this section, as the results are exact, we present the numerical approximations using double precision accuracy. Then,

$$r_1^2 = |U - V|^2 = a \, S_1^2 + b \, S_2^2 + c S_1 S_2, \quad r_2^2 = |U - W|^2 = A \, S_1^2 + B \, S_2^2 + C S_1 S_2. \tag{100}$$

where (again, only writing out the numerical approximation of the complicated exact expression),

$$a \approx 3.914892813645649, \quad b \approx 5.140576474687263, \quad c \approx 8.97213595499958,$$
$$A \approx 6.19959346906221, \quad B \approx 3.246149283687347, \quad C \approx -8.972135954999578. \tag{101}$$

As shown in the Appendix A, this can be solved to obtain

$$S_1 = \sqrt{\frac{-K_1 \pm \sqrt{K_1^2 - 4K_2 K_0}}{2K_2}}, \qquad\qquad S_2 = \sqrt{\frac{\Delta_r - \Delta_a S_1^2}{\Delta_b}}, \tag{102}$$

where

$$K_0 = r_1^4 + \frac{b^2 \Delta_r^2}{\Delta_b^2} - 2\frac{b r_1^2 \Delta_r}{\Delta_b},$$

$$K_1 = 2\frac{b r_1^2 \Delta_a}{\Delta_b} - 2b^2\frac{\Delta_r \Delta_a}{\Delta_b^2} - 2ar_1^2 + (2\,a\,b - c^2)\frac{\Delta_r}{\Delta_b},$$

$$K_2 = a^2 + \frac{b^2 \Delta_a^2}{\Delta_b^2} + (c^2 - 2\,a\,b)\frac{\Delta_a}{\Delta_b},$$

$$\Delta_a = a\,C - A\,c, \quad \Delta_b = b\,C - B\,c, \quad \Delta_r = r_1^2\,C - r_2^2\,c. \tag{103}$$

This leads to the regular p-cages listed in Table 5.

**Table 5.** List of all regular DArt p-cages with $a = b = c = (P_1 - 3)/3$ and $A = B = C = D = E(P_2 - 5)/5$.

| $P_1$ | $P_2$ | $S_1$ | $S_2$ | $S_2/S_1$ | $P_1$ | $P_2$ | $S_1$ | $S_2$ | $S_2/S_1$ |
|---|---|---|---|---|---|---|---|---|---|
| 6 | 10 | 2.427051 | 2.500000 | 1.030057 | 12 | 15 | 4.179611 | 4.470477 | 1.069592 |
| 6 | 15 | 3.313585 | 3.273659 | 0.987951 | 12 | 20 | 5.056432 | 5.235659 | 1.035445 |
| 6 | 20 | 4.190407 | 4.038842 | 0.963830 | 15 | 15 | 4.600750 | 5.052477 | 1.098185 |
| 9 | 10 | 2.866744 | 3.107640 | 1.084031 | 15 | 20 | 5.477571 | 5.817659 | 1.062087 |
| 9 | 15 | 3.753278 | 3.881299 | 1.034109 | 18 | 15 | 5.019322 | 5.630930 | 1.121851 |
| 9 | 20 | 4.630099 | 4.646482 | 1.003538 | 18 | 20 | 5.896144 | 6.396112 | 1.084796 |
| 12 | 10 | 3.293076 | 3.696818 | 1.122603 | 21 | 20 | 6.313258 | 6.972550 | 1.104430 |

## 8. Results Summary and Conclusions

We identified six types of geometries from which bi-homogenoues symmetric 1-2 p-cages can be constructed, each corresponding to planar graphs. Each planar graph corresponds to a hole polyhedron, which is one of the following: a prism with an even number of edges (including the cube), the truncated octahedron, the truncated cuboctahedron, the truncated icosidodecahedron, the rhombic dodecahedron, or the rhombic triacontahedron. The p-cages we found for the different holes–polyhedra are given in Table 6.

We were able to construct regular BiHS12 p-cages for all of them except for the truncated icosidodecahedron, which does not have any. We present images of some of the regular p-cages in Figure 13 and of some of the best near-miss p-cages for each type of hole–polyhedron, except for DArd and DArt (for which, by symmetry, all the p-cages are regular), in Figure 14.

The full list of all BiHS12 p-cages with deformations below 10% is given in the Supplementary Material. A picture of all regular BiHS12 p-cages as well as pictures of all near-miss BiHS12 p-cages with deformations below 1% are also provided there.

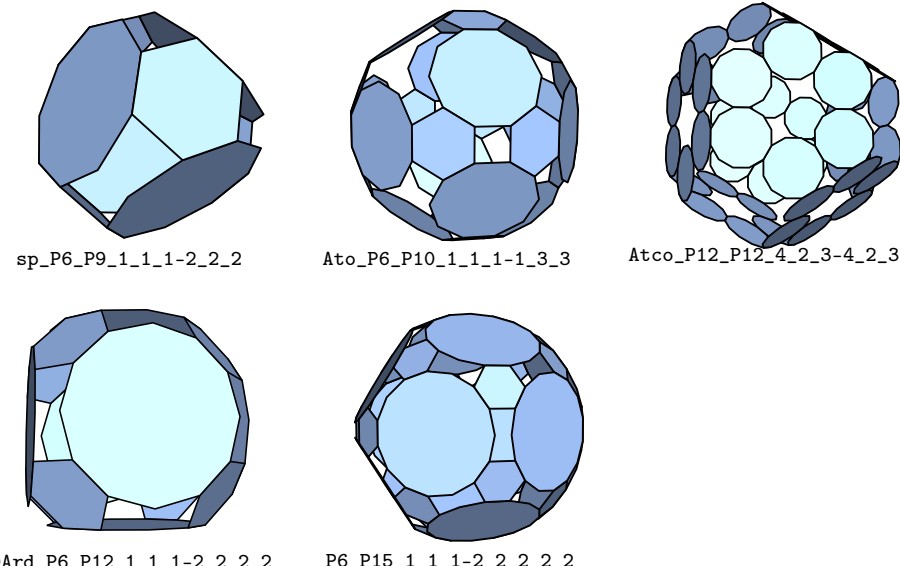

sp_P6_P9_1_1_1-2_2_2    Ato_P6_P10_1_1_1-1_3_3    Atco_P12_P12_4_2_3-4_2_3

DArd_P6_P12_1_1_1-2_2_2_2    P6_P15_1_1_1-2_2_2_2_2

**Figure 13.** Graphic representation of some regular p-cages.

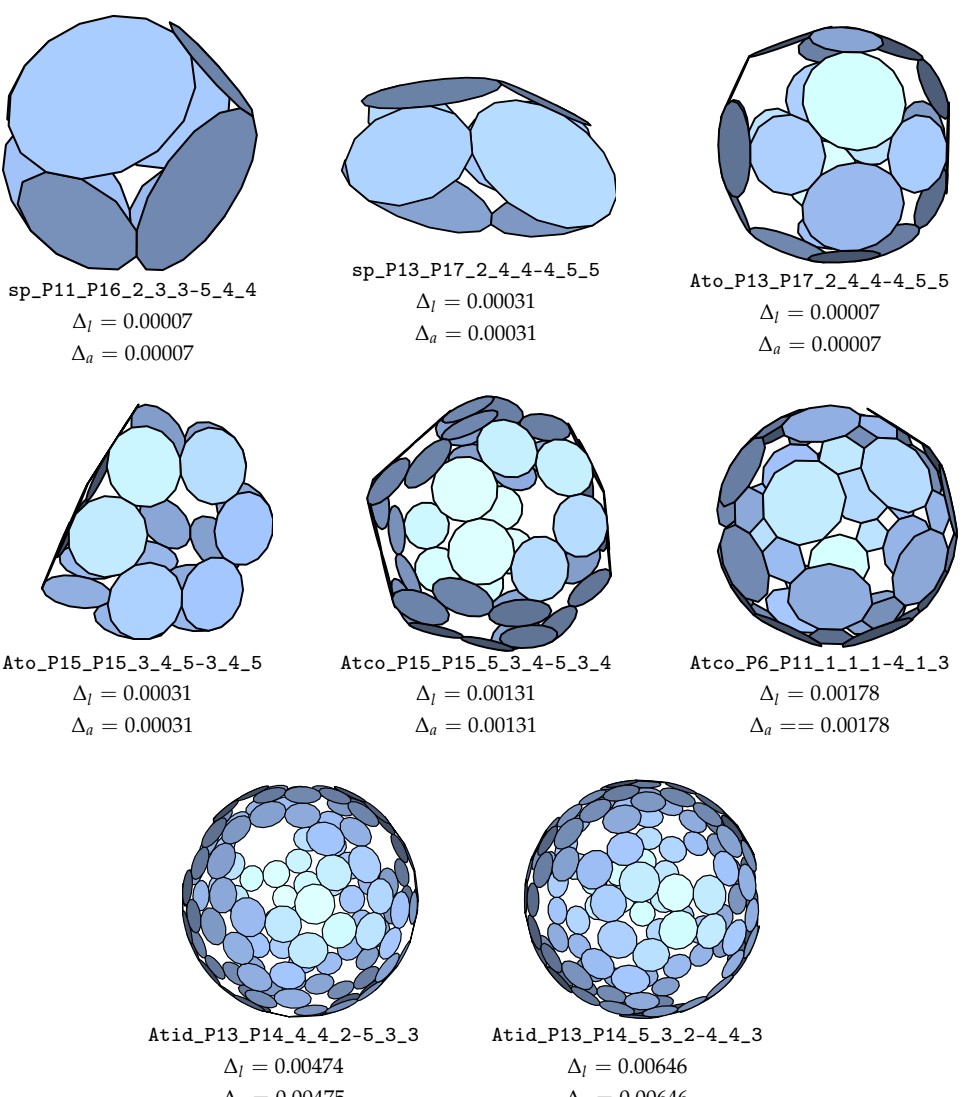

**Figure 14.** Graphic representation of some near-miss p-cages.

**Table 6.** Number of p-cages found for each type of hole–polyhedron graph.

| Type | Regular | $0 < \Delta \leq 1\%$ | $1\% < \Delta < 10\%$ |
| --- | --- | --- | --- |
| sp | 28 | 49 | 556 |
| Ato | 11 | 24 | 987 |
| Atco | 1 | 18 | 1512 |
| Atid | 0 | 11 | 950 |
| DArd | 19 | 0 | 0 |
| DArt | 14 | 0 | 0 |

**Supplementary Materials:** The following supporting information can be downloaded at: https://www.mdpi.com/article/10.3390/sym15091804/s1, BiHS12_tables.pdf: full list of all the BiHS12 p-cages with a deformation below 10%; Regular_pcages.pdf: picture of each regular BiHS12 p-cage; Best_BiHS12_cages.pdf: picture of every near-miss BiHS12 p-cage with a deformation below 1%.

**Author Contributions:** B.P. derived the methods and wrote the software to identify the bi-homogeneous-symmetric-1-2 planar graph, the regular bi-homogeneous-symmetric-1-2 p-cages, and the near-miss bi-homogeneous-symmetric-1-2 p-cages. Á.L. Checked the analytic derivations. Conceptualization, B.P.; formal analysis, B.P; funding acquisition B.P; investigation, B.P., Á.L.; methodology, B.P.; resources, B.P.; software, B.P.; writing—original draft, B.P. and Á.L. All authors have read and agreed to the published version of the manuscript.

**Funding:** This study was funded by the Leverhulme Trust Research Project grant RPG-2020-306.

**Data Availability Statement:** The software used to determine all the bi-homogeneous-symmetric-1-2 planar graph as well as that used to determine the coordinates of the least deformed p-cages are both available from zenodo: https://doi.org/10.5281/zenodo.8252395. The coordinates of all the near-miss p-cages with deformations below 10% are available as off files from zenodo: https://doi.org/10.5281/zenodo.8252453.

**Acknowledgments:** The p-cage figures were generated using geomview from www.geomview.org (accessed on 12 September 2023).

**Conflicts of Interest:** The authors declare no conflict of interest.

## Abbreviations

The following abbreviations are used in this manuscript:

| | |
|---|---|
| TRAP | trp RNA-binding attenuation protein. |
| RNA | Ribonucleic acid: a nucleic acid present in all living cells. |
| DNA | Deoxyribonucleic acid: a nucleic acid present in all living cells. |

## Appendix A. Solving a Pair of Quadratic Algebraic Equations

To compute the coordinates of the vertices of the regular `DArt` p-cages, we need to solve

$$r_1^2 = a\,S_1^2 + b\,S_2^2 + cS_1S_2, \qquad\qquad r_2^2 = A\,S_1^2 + B\,S_2^2 + CS_1S_2, \qquad (A1)$$

where the coefficients $a$, $A$, $b$, $B$, $c$, and $C$ take explicit values for each type of p-cage, and $r_1$ and $r_2$ are determined, respectively, from $P_1$ and $P_2$ by (26). We can determine the values of the scaling parameters $S_1$ and $S_2$ by computing

$$S_1\,S_2 = \frac{r_1^2 - a\,S_1^2 - b\,S_2^2}{c} = \frac{r_2^2 - A\,S_1^2 - B\,S_2^2}{C},$$

$$S_2^2\left(\frac{b}{c} - \frac{B}{C}\right) = \frac{r_1^2}{c} - \frac{r_2^2}{C} + S_1^2\left(\frac{A}{C} - \frac{a}{c}\right),$$

$$S_2^2 = \frac{(r_1^2\,C - r_2^2\,c) + S_1^2(A\,c - C\,a)}{b\,C - B\,c}. \qquad (A2)$$

Squaring the first equation and expanding the squared sums yield

$$c^2\,S_1^2\,S_2^2 = \left(r_1^2 - a\,S_1^2 - b\,S_2^2\right)^2$$
$$= r_1^4 + a^2\,S_1^4 + b^2\,S_2^4 - 2ar_1^2S_1^2 - 2br_1^2S_2^2 + 2abS_1^2S_2^2. \qquad (A3)$$

Defining

$$\Delta_a = a\,C - A\,c, \qquad\qquad \Delta_b = b\,C - B\,c, \qquad\qquad \Delta_r = r_1^2\,C - r_2^2\,c, \qquad (A4)$$

and substituting (A2) into (A3), we have

$$c^2 S_1^2 \left( \frac{\Delta_r - \Delta_a S_1^2}{\Delta_b} \right) = r_1^4 + a^2 S_1^4 + b^2 \left( \frac{\Delta_r - \Delta_a S_1^2}{\Delta_b} \right)^2 - 2ar_1^2 S_1^2$$
$$- 2br_1^2 \left( \frac{\Delta_r - \Delta_a S_1^2}{\Delta_b} \right) + 2abS_1^2 \left( \frac{\Delta_r - \Delta_a S_1^2}{\Delta_b} \right). \tag{A5}$$

As a result, we can write

$$0 = K_2 S_1^4 + K_1 S_1^2 + K_0 \tag{A6}$$

where

$$K_2 = a^2 + \frac{b^2 \Delta_a^2}{\Delta_b^2} + (c^2 - 2\,a\,b) \frac{\Delta_a}{\Delta_b},$$
$$K_1 = 2 \frac{br_1^2 \Delta_a}{\Delta_b} - 2b^2 \frac{\Delta_r \Delta_a}{\Delta_b^2} - 2ar_1^2 sp_r eg.txt + (2\,a\,b - c^2) \frac{\Delta_r}{\Delta_b},$$
$$K_0 = r_1^4 + \frac{b^2 \Delta_r^2}{\Delta_b^2} - 2 \frac{br_1^2 \Delta_r}{\Delta_b}. \tag{A7}$$

Finally,

$$S_1 = \sqrt{\frac{-K_1 \pm \sqrt{K_1^2 - 4K_2 K_0}}{2K_2}},$$
$$S_2 = \sqrt{\frac{\Delta_r - \Delta_a S_1^2}{\Delta_b}}. \tag{A8}$$

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
