# Peer review of "Near-Miss Bi-Homogenous Symmetric Polyhedral Cages"

_symmetry, doi:10.3390/sym15091804_

Round 1

Reviewer 1 Report

PFA

Moderate editing of English language required

Author Response

See attached file. Notice that all the changes have been coloured in red in the revised manuscript.

Reviewer 2 Report

The manuscript covers a very interesting topic of novel cage structures, which are less regular than regular polyhedrons, but add a lot of potential symmetrical arrangements of atoms or molecules. The presentation is very neat, the paper is well written, literature adequately cited, figures nicely done, including a meaningful use of color, almost no typos/errors (Figure 7 caption: underlining or underlying? Table 7/8 should read Figure 13/14, please check also the references to them in the text, the third picture in the first row seems to be wrongly truncated? Results should better read Conclusions), maybe some of the more detailed calculations or parts like appendix 1 could be outsourced to another supplemental material, and maybe the stated numerical precision could be given another thought (don't 6 decimal digits always suffice?), but overall it is a paper, how it should be submitted. I thus can recommend publication of the manuscript after minor revision.

Author Response

See attached file. Notice that all the changes have been coloured in red in the manuscript.
